

# A high-resolution air temperature data set for the Chinese Tianshan Mountains in 1979-2016

Lu Gao[1,2,3,4], Jianhui Wei[5], Lingxiao Wang[6], Matthias Bernhardt[7], Karsten Schulz[7], Xingwei Chen[1,2,3,4]

[1] Fujian Provincial Engineering Research Center for Monitoring and Assessing Terrestrial Disasters, Fujian Normal University, Fuzhou, 350007, China
[2] Institute of Geography, Fujian Normal University, Fuzhou, 350007, China
[3] College of Geographical Science, Fujian Normal University, Fuzhou, 350007, China
[4] State Key Laboratory of Subtropical Mountain Ecology (Funded by Ministry of Science and Technology and Fujian Province), Fujian Normal University, Fuzhou, 350007, China
[5] Institute of Meteorology and Climate Research (IMK-IFU), Karlsruhe Institute of Technology, Campus Alpine, Garmisch-Partenkirchen, Germany
[6] Department of Geography, Ludwig-Maximilians-Universität München, Munich, 80333, Germany
[7] Institute of Water Management, Hydrology and Hydraulic Engineering, University of Natural Resources and Life Sciences, Vienna, Vienna, 1190, Austria

*Correspondence to*: Lu Gao (l.gao@foxmail.com)

**Abstract.** The Chinese Tianshan Mountains has a complex ecological environment system. It not only has a large number of desert oases, but also gave birth to a large number of glaciers. The arid climate and the shortage of water resources are the important factors to restrict the socio-economic development in this area. This study presents a unique high-resolution (1 km, 6-hourly) air temperature data set for the Chinese Tianshan Mountains (41.1814-45.9945 °N, 77.3484-96.9989 °E) from 1979 to 2016 based on a robust statistical downscaling framework. The data set was validated by 24 meteorological stations at daily scale. Compared with original ERA-Interim temperature, the Nash-Sutcliffe efficiency coefficient increased from 0.90 to 0.94 over all test sites. Around 24% of root-mean-square error was reduced from 3.75 to 2.85 °C. A skill score based on the probability density function, which was used to validate the reliability of the new data set for capturing the distributions, enhanced from 0.86 to 0.91 for all test sites. We conclude that the new high-resolution data set is reliable for climate change investigation over the Chinese Tianshan Mountains. This data set would be helpful for the potential users for better local climate monitoring, modelling and environmental studies in the Chinese Tianshan Mountains. The data set presented in this article is published in Network Common Data Form (NetCDF) at doi: 10.1594/PANGAEA.887700. The data set includes 288 nc files and one user guidance in txt file.

# 1 Introduction

The near surface air temperature is the primary indicator for climate change, which significant impacts on the local as well as the global water, energy and matter cycle (e.g. Bolstad et al., 1998; Gao et al., 2012; Gao et al., 2014a; Prince et al., 1998). Air temperature is a necessary input variable for most of the hydrological and environmental models, because it controls a large variety of environmental processes. Long-term and high-resolution temperature including historic, current and future



series is the prerequisite for accurate climate change assessment especially on a regional scale (Gao et al., 2012; Pepin and Seidel, 2005; Minder et al., 2010; Maurer et al., 2002; Mooney et al., 2011). However, as the most common sources for air temperature time series, observational networks suffer from the low-station density in complex terrains, in particular at high mountains (Gao et al., 2014a). The installation and maintenance of the stations in these regions are the main challenges

(Kunkel, 1989; Rolland, 2003). In order to obtain spatially continuous temperature data, interpolation technologies like natural neighbourhood, inverse distance weighting and a series of Kriging methods are usually applied. However, these methods rely on the density of surface stations. The reliability of interpolated results decreases with increasing distance from the selected stations, especially for the inverse distance weighting method. Thus, interpolation approaches may induce large errors for a large region with low density of stations (e.g. Vogt et al., 1997).

In contrast to low availability of observations, reanalysis products provide long-term and spatially consistent data sets, which have been increasingly applied for climate change assessment in the past two decades (e.g. Gao et al., 2014a; Mooney et al., 2011). Reanalysis is designed to estimate the state of real atmosphere and land surface characteristics by assimilating a lot of observations (Decker et al., 2012; Simmons et al., 2010). However, reanalysis contains uncertainties, such as observation changes and model misrepresentation (Simmons et al., 2010). With the development of reanalysis assimilation system, the

spatial resolution has been enhanced down to 0.125°, for example ERA-Interim.. However, the local processes such as temperature inversion in deep valleys, as well as snowpack accumulation and melting, are still not explicitly considered. Furthermore, due to the heterogeneity over the land surface, many hydrological and climatic impact models go for applications on high resolutions, which tend to run on the scale of 0.1-1 km (Bernhardt and Schulz, 2010; Gao et al., 2012; Maraun et al., 2010). To this end, downscaling and correcting reanalysis data is necessary (Gao et al., 2012, 2014b, 2016).

Previous studies have shown that the elevation difference between the reanalysis grid point and the corresponding meteorological station leads to a large systematic bias (Gao et al., 2012, 2014a, 2016). Thus, the elevation correction scheme based on a lapse rate, which explains the empirical relationship between air temperature and altitude, can reduce this bias significantly. A constant lapse rate from the range of -6.0 °C km$^{-1}$ to -6.5 °C km$^{-1}$ (e.g. Dodson and Marks, 1997; Lundquist and Cayan, 2007; Maurer et al., 2002; Marshall et al., 2007) is commonly used. The monthly temperature gradients within

the atmosphere are also widely applied in different regions (Kunkel, 1989; Liston and Elder, 2006). However, previous studies have shown that a fixed lapse rate may be problematic since the values of the lapse rate can vary significantly within short time periods of less than a month (Minder et al., 2010; Lundquist and Cayan, 2007; Rolland, 2003). To tackle this issue, using the lapse rate calculated from the meteorological stations has the best performance in many regions (e.g. Gao et al., 2012, 2017). However, the observed lapse rate completely relies on the density of sites and it is not applicable for the regions

without observations like high mountains.

Gao et al. (2012) introduced one another strategy that obtains the temporal variability of lapse rates by using ERA-Interim internal temperature profiles. It is completely derived from ERA-Interim on the internal pressure levels. It is therefore



independent of local station observations. This method has been successfully applied for the cases in the European Alps and the Tibetan Plateau (Gao et al., 2012, 2017). Furthermore, this approach has the potential to be used to downscale ERA-Interim temperature data for any other high mountainous areas. Following this approach, for the first time, the 0.25°×0.25°, 6-hourly ERA-Interim 2 m temperature data from 1979 to 2016 is downscaled to 1 km, 6-hourly for the Chinese Tianshan

Mountains (CTM). The temperature data set presented here is extraordinarily unique, because it covers such a large, complex terrain area with a long-term continuous both in space and in time. The validation with observations from meteorological stations shows that this data set is generally reliable and suitable for climate change impact assessment as well as for hydrological and environmental modelling.

The specific information about the Chinese Tianshan Mountains is described in Sect. 2. The used data including ERA-

Interim and observations, downscaling methods as well as evaluation criteria are presented in Sect. 3. Section 4 provides the validation results, and discussion and conclusions are presented in Sect. 5. Data accessibility is presented in Sect. 6.

## 2 Study area

The Tianshan Mountains is one of the seven major as well as the largest independent latitude mountain systems over the world. It spans four countries including China, Kazakhstan, Kyrgyzstan, and Uzbekistan in the east-west direction. It

stretches around 2500 km length with an average width of 250-350 km (Hu, 2004, Chen et al., 2014; Wang et al., 2011). The East Tianshan Mountains is called the Chinese Tianshan Mountains (CTM) which extends over 1700 km in Xinjiang Province, China (Figure 1). It occupies approximately one third of the entire area of Xinjiang Province (~570,000 km$^2$). The CTM consists of three parallel mountains (Hu, 2004; Wang et al., 2011). The north branch of the Mountains mainly includes the Ala Mount, Keguqin, Borokoonu, and Bogda Mountain. The central branch includes the Arakar, Nalati, Erwingen, and

Hora Mountain. The south branch mainly includes Kochsal, Khark, Lerke, and Karake Mountain (Hu, 2004). The elevation decreases from west to east with an average elevation of around 4000 m. The CTM also is a boundary for the northern and southern hillsides, which is represented by the Junggar Basin and the Tarim Basin, respectively. Mt. Tomur is the highest peak of the Tianshan Mountains with the elevation of 7443.8 m (Hu, 2004).

The Tianshan Mountains has its special arid climate regime due to the longest distance to the sea compared with other major

mountain systems over the world. Many rivers such as the Sir River, the Chu River and the Yili River are originated in the Tianshan Mountains (Chen et al., 2014; Chen et al., 2017; Hu, 2004). The CTM not only has a large number of desert oases in the basin, but also gave birth to a large number of glaciers. In statistics, there are 9035 glaciers in the CTM with an area of 9225 km$^2$ and a volume of 1011 km$^3$ (Shi, 2008; Shi et al., 2010). However, most of the glaciers in CTM are in a state of rapid retreat due to the global warming (e.g. Ding et al., 2006; Li et al., 2003; Li et al., 2010). As a climate transition zone,

the CTM is known as the "wet island" in arid regions (Chen et al., 2015; Deng and Chen, 2017). The glaciers and snow cover in high mountains are the most sensitive indicators of climate change. Global warming, especially the significantly increased



temperature impacts on the retreat of glaciers and the ablation of snow, which further influences the regional water resources, and ecological environment (Chen et al., 2014; Chen et al., 2017; Wang et al., 2011; Wei et al., 2008; Zhang et al., 2012).

Because of the special geographical location and complex terrain, the temperature changes in northern and southern hillsides of the CTM are significant in regional and seasonal. Due to limited long-term observations in the CTM, the link between

climate change and glacier variation is still unclear. In the past decades, most of the studies have focused on the Glacier No.1 at the headwaters of Urumqi River (Li et al., 2003). With the development of Geographic Information System (GIS), Remote Sensing (RS) and climate reconstruction technologies, great progress has been made on the glacier fluctuation investigations (e.g. Chen et al., 2012; Li et al., 2010). However, a reliable long-term series of temperature data is great needed to explore the glacier variations under the warming and wetting climate (Chen et al., 2015).

**3 Data and method**

**3.1 ERA-Interim data**

The European Centre for Medium Range Weather Forecast (ECMWF) reanalysis product ERA-Interim is downscaled in this study. ERA-Interim provides data from 1979 onwards, and continues in real time (Berrisford et al., 2009; Dee et al., 2011). The Cycle 31r2 of ECMWF's Integrated Forecast System (IFS) was used for the ERA-Interim product. Compared with

ERA-40, ERA-Interim improved significant in the representation of the hydrological cycle, the quality of the stratospheric circulation, as well as the handling of biases by using a 4-dimensional variation analysis (Dee and Uppala, 2009; Simmons et al., 2006; Uppala et al., 2008; Dee et al., 2011). ERA-Interim model in this configuration comprises 60 vertical levels, with the top level at 0.1 hPa. It used the T255 spectral harmonic representation for the basic dynamical fields and a reduced Gaussian grid (N128) with an approximately uniform spacing of 79 km (Dee et al., 2011; Uppala et al., 2008). ERA-Interim

assimilates four analyses per day at 00, 06, 12 and 18 UTC. Two 10-day forecasts with a 3-hour resolution are initialized on the basis of the 00 and 12 UTC analyses (Dee et al., 2011; Uppala et al., 2008). Due to usage limitation, 6-hourly assimilation data at 00, 06, 12 and 18 UTC from 1979-2016 which are projected on a grid of 0.25°×0.25° were used. This grid is interpolated from the original reduced Gaussian grid. The used output variables are 2 m temperature, surface pressure, as well as temperature and geopotential height at 925 hPa, 850 hPa, 700 hPa, 600 hPa and 500 hPa levels. The geopotential

height is related to the variation of gravity with latitude and elevation, and is calculated by the normalization of the geopotential over the gravity (Gao et al., 2012).

**3.2 Observations**

Daily mean air temperatures from 24 meteorological stations located on the CTM derived from the China Meteorological Data Sharing Service System of the National Meteorological Information Center (CMA-CMDC, http://data.cma.cn/) are

used in the analysis. An overview of the observations is given in Figure 1 and Table 1. Some stations were relocated which



lead to the different initial time (Table 1). The elevation was adjusted for the station No. 24 but the coordinate is not changed. The quality of observed data is strictly controlled by the National Meteorological Information Center of China. The observed daily mean temperature is averaged from four records (6-hourly) from previous 20:00 to 20:00 in Beijing time. Therefore, 6-hourly ERA-Interim temperature is adjusted according to the time differences to match observations.  Please note that, only 7

out of 24 meteorological stations are available for international users via CMA-CMDC with a registration. These 7 sites are a small part of data set, which covers 194 sites since January 1951 over China for global exchange (http://data.cma.cn/en/?r=data/detail&dataCode=SURF_CLI_CHN_MUL_DAY_CES_V3.0). These 7 sites are Nos. 2, 5, 6, 8, 15, 18 and 23 in this study.

### 3.3 Downscaling method

The original ERA-Interim temperature can be downscaled via Equation (1). $T_{ERA\_2m}$ is the original 6-hourly ERA-Interim 2 m temperature at 0.25°×0.25° grid. Γ describes the lapse rate with a decrease of air temperature with elevation. $\Delta h$ is the altitude difference between 1 km grid (re-sampled from 90 m of SRTM DEM to 1 km) and ERA-Interim grid.

$$T_i = T_{ERA\_2m} + \Gamma \times \Delta h \tag{1}$$

Here, Γ represents the ERA-Interim internal lapse rates calculated from temperatures and geopotential heights at different

pressure levels such as between 500 hPa and 700 hPa. In general, pressure levels covered the largest altitude range according to each 1 km grid are used. More details about the downscaling method can be found in Gao et al. (2012, 2017).

### 3.4 Evaluation criteria

In order to evaluate the downscaled results, two statistical accuracy measures are applied. The root mean square error (RMSE) is used for an assessment of the bias between downscaled data set and observations (Equation (2)). The Nash-

Sutcliffe efficiency coefficient (NSE) evaluates the performance of the downscaled data set using Equation (3), which ranges from 1 (perfect fit) to minus infinity (worst fit) (Nash and Sutcliffe, 1970).

$$RMSE = \sqrt{\frac{1}{n}\sum_{t=1}^{n}(T_o - T_d)^2} \tag{2}$$

$$NSE = 1 - \frac{\sum_{t=1}^{n}(T_o - T_d)^2}{\sum_{t=1}^{n}(T_o - \overline{T_o})^2} \tag{3}$$

Where, $T_o$ is observed temperature at time $t$, $T_d$ is downscaled temperature at time $t$, and $n$ is number of records in the same

time series.

A measure of skill based on probability density function (PDF) proposed by Perkins et al. (2007) was applied in this analysis. This skill score calculates the cumulative minimum intersections (or overlaps) between the two distribution binned values in the given PDFs. The skill score ranges from 0 to 1. The value 1 means the PDF shape of downscaled temperature match the observed PDF perfectly. The value 0 means there is no common area between the PDFs of observations and downscaled



temperatures. In other words, these two PDFs are independent completely. The PDF-based skill score is calculated via Equations (4) and (5):

$$\text{skill score} = \sum_{1}^{m} P_m \tag{4}$$

$$P_m = \begin{cases} P_d^m, & P_d^m < P_o^m \\ P_o^m, & P_d^m \geq P_o^m \end{cases} \quad (m = 1,2,3 \dots n) \tag{5}$$

Where n is the number of bins for PDF calculation. $P_d$ is the frequency of values in a given bin from the downscaled temperatures. $P_o$ is the frequency of values in a given bin from the observations (Perkins et al., 2007, Gao et al., 2016). In this downscaling, 1 °C is used as the intervals of bins for all skill score calculations. The PDF-based evaluation method allows merging data from multiple stations across different time periods (Perkins et al., 2007, Gao et al., 2016). This evaluation method has been proven that it can show more credible climate variations especially for extreme values than the
conventional mean-based assessment method (Gao et al., 2016; Mao et al., 2010).

Furthermore, 1% quantile and 5% quantile temperatures are represented for extreme low temperatures, while 95% quantile and 99% quantile temperatures are selected for extreme high temperatures. Quantile function is more reliable than the absolute minimum and maximum values, especially for the data sets in different time series. It is worth noting that downscaled temperature is accordingly averaged over 9 grid points surrounding each meteorological station. The
downscaled temperature is in 1979-2016 and the observation is in 1979-2013. These different durations of two data sets does not affect the comparison using PDF-based skill score and quantile function.

## 4 Results

### 4.1 Evaluation of original ERA-Interim temperature data

The correlation coefficient ranges from 0.949 to 0.995 with an average of 0.986 for all stations (the detailed results not
shown here). This high correlation coefficient indicates that the original ERA-Interim temperature captures the temporal variation of observations very well. Table 2 shows the comparison of original ERA-Interim 2 m temperature with daily observations from 24 meteorological stations. The NSE ranges from 0.46 to 0.97 for all stations. Only five stations (Nos. 6, 8, 9, 13 and 15) have the NSEs lower than 0.90. The average NSE of 0.90 for all stations reveals that ERA-Interim reproduces the observations very well. The lowest NSE is found at station No. 9 (Baluntai), while the highest NSE is found at No. 5
(Qitai) and No. 24 (Hongliuhe). The RMSE ranges from 2.05 to 7.76 °C with an average of 3.75 °C for all stations. Three stations (Nos. 9, 13 and 15) have RMSEs higher than 5 °C. As NSE indicted, station No. 9 (Baluntai) has the largest RMSE (7.76 °C), followed by station No. 15 (7.69 °C). The smallest RMSE is found at No. 24 (Hongliuhe).

Station No. 9 (Baluntai) is located in a valley with an elevation of ~1700 m, while the terrain height of the corresponding grid in ERA-Interim is 2616 m (Table 1). The around 900 m elevation difference may be responsible for the large RMSE.



Station No. 13 (Bayinbrook) locates in the hinterland of the CTM and in the Kaidu River valley. The elevation difference between site and ERA-Interim grid is 383 m, which partly accounts for the large RMSE. Station No. 15 (Turfan) locates in the basin in the southern hillside of the CTM with the elevation of only 35 m. The ERA-Interim grid height (1115 m) is much higher than the site, which may lead to the large bias. Figure 2 shows the comparison of observation and original

ERA-Interim temperature data at station Nos.10, 24, 9 and 15. ERA-Interim underestimates observations for station Nos. 9 and 15 due to its higher grid elevation, especially for higher temperatures (Figure 2c and 2d). The ERA-Interim estimates quite well at station Nos. 10 and 24 for both lower and higher temperatures (Figure 2a and 2b).

The PDF-based skill score ranges from 0.67 to 0.94 with an average of 0.86 over all stations. Four stations (Nos. 8, 9, 13 and 15) have the skill scores lower than 0.8. The highest skill score is found at station No. 24 while the smallest one is found at

station No. 9. Figure 3 shows greater detail especially in terms of the temperature bins of the PDFs, which could help to easily identify how well ERA-Interim estimates for lower and higher temperatures. For station No. 10, the original ERA-Interim captures the shape of the observed PDF very well, particularly in the range of 0 °-15 °C. In the range of round -10 °-0 °C and higher than 20 °C, the observed probability is higher than original ERA-Interim. However, the probability of original ERA-Interim is higher than observations for lower temperatures (< -10 °C). For station No. 24, the original ERA-

Interim fits the shape of observed PDF very well almost for entire temperature range. It only has lower probability for temperatures lower than -10 °C. For higher temperature (> 20 °C), the probability of original ERA-Interim is slight higher than observation. For station No. 9, the shape of PDF from original ERA-Interim is higher (higher probability) for the temperatures lower than 12 °C and is lower (lower probability) for the temperatures higher than 12 °C compared with observed PDF. For station No. 15, the original ERA-Interim does not match the shape of observed PDF generally. The

probability of original ERA-Interim is much higher than observation for the temperatures cooler than about 25 °C while is much lower than observation for higher temperatures (> 25 °C). The above analysis indicates that although the original ERA-Interim captures the temporal variation of observation very well, the large RMSE and poor performance for extreme temperatures suggested that a downscaling process of ERA-Interim is necessary for the application at individual sites.

### 4.2 Validation of downscaled ERA-Interim temperature data

The average correlation coefficient changes from 0.986 to 0.987 for all stations (not shown here) with respect to original and downscaled ERA-Interim temperature. Table 2 shows the comparison of downscaled ERA-Interim 2 m temperature with daily observations from 24 meteorological stations. The NSE for downscaled ERA-Interim ranges from 0.69 to 0.98 for all sites. The average NSE enhances by 5% from 0.90 to 0.94. The NSE significantly increases from 0.46 to 0.82 for station No. 9 and from 0.71 to 0.94 for station No. 15, respectively. Applying the downscaling framework does not lead to an increased

NSE at stations Nos. 2, 4, 5 and 13, especially for the station No. 13 (Bayinbrook). It indicates that other factors rather than altitude have higher influence on temperature changes much greater than altitude. The RMSE of downscaled ERA-Interim changes from 1.53 to 7.80 °C. An average 0.9 °C (24%) of RMSE is reduced from 3.75 to 2.85 °C. The RMSE reduces at all



sites except Nos. 2, 4, 5, 13 and 16. The RMSE increases around 0.5 °C for station Nos. 2, 4, 5 and 16. 1.15 °C of RMSE increased for station Nos. 13. 42% and 55% of RMSE is reduced for station No. 9 and No. 15, respectively. 16 out of 24 sites have RMSEs lower than 3.0 °C after downscaling process. Figure 4 shows the comparison of observation and downscaled ERA-Interim temperature data at station Nos.10, 24, 9 and 15. The scatter plot shows slight improvement (for higher

temperature) at station Nos. 10 and 24 compared with original ERA-Interim (Figure 2 and 4). However, the scatter plot are more concentrated along the 1:1 line for station Nos. 9 and 15, especially No. 15 (Figure 4c and 4d), which suggests that the downscaling method works significant at these two stations.

The average PDF-based skill score increases from 0.86 to 0.91 for all stations (Table 2). The downscaled results show better probability distribution functions compared with original ERA-Interim for all stations, except No. 16 (Baicheng). Although

five stations have increased RMSEs, the PDF-based skill score enhances at four of five stations (station Nos. 2, 4, 5, and 13). It indicates that the downscaling procedure reduces the PDF discrepancy between original ERA-Interim and observation. Figure 3 shows more details on the performance of downscaling for four representative stations. For station No. 10, the PDF curve of downscaled ERA-Interim shifts right compared with the original one and it fits perfect with observed PDF in the range of temperature lower than -8 °C. The probability of downscaled ERA-Interim is smaller than observation in the range of -8 °-

22 °C (Figure 3a). For station No. 24, the PDF shape of downscaled ERA-Interim shifts slight left against the original one and it fits the observed PDF much better for temperatures higher than 0 °C (Figure 3b). For station No. 9, the probability of downscaled ERA-Interim is much higher than observed temperature lower than 15 °C, especially in the range of 0 °-15 °C. It performs better than the original ERA-Interim for higher temperatures (> 15 °C). A much better agreement between the PDF curves of downscaled ERA-Interim and observation is found at station No. 15 (Figure 3d). The probability of downscaled

ERA-Interim is much close to observation compared with original ERA-Interim in the range of 15° -35 °C. The downscaling procedure reduces significant PDF discrepancy for station No. 15 which has a lower elevation (35 m). The validation suggests that the downscaling method is reliable generally and it could reduce RMSE and PDF discrepancy significantly for most of the stations, although it does not perform well for some individual sites.

### 4.3 Climatology of the Chinese Tianshan Mountainous based on the high-resolution data set

The low density of meteorological stations in the CTM may lead to an uncertainty for the representation of the entire area temperature climatology. Interpolation is also risk to represent the climatology because it is based on the meteorological stations. Here, the entire area climatology is investigated using the downscaled high-resolution data set from 1979-2016. Quantile temperature is more reliable than absolute minimum and maximum temperatures if outlier exists. Thus, 1% quantile and 99% quantile of long-term series is selected to represent the extreme cold and hot temperatures, respectively. Figure 5

shows a new comprehensive climatology of extreme low temperature over the CTM based on downscaled ERA-Interim temperature. The 1% quantile temperature ranges from -49 to -12 °C, which is consistent with the topography. The lower temperatures could be found at the Borokonou, Bogda and Khark Mountain. The extreme cold temperatures (< -40 °C) are in



Tomur Peak, Bogda Peak, and Tianger Peak. The higher temperatures could be found at the Yili River Valley, Junggar Basin, Turfan Basin, Hami Basin and Tarim Basin. The extreme hot temperatures are found at the border of the Yili River Valley and the west of Tarim Basin.

Figure 6 shows the 99% quantile temperature which represents the extreme warm temperature in the CTM. The Turfan Basin
and Hami Basin are the hottest area over the entire CTM. The highest temperature could be 45 °C. The temperature in the high mountain areas such as Tomur Peak and Bogda Peak could be higher than 0 °C. The minimum temperature of 99% quantile is around -3 °C. Figure 7 shows the mean temperature in the CTM. The temperature ranges from -25 to 16 °C. The mean temperature distribution is consistent with extreme cold and hot temperatures. It suggests that the topography is a significant impact factor for temperature.

Table 3 illustrates the climatology comparison for the entire CTM using downscaled ERA-Interim temperature and 24 stations. 1% and 5% quantile are selected to represent the colder temperature while 95% and 99% quantile are for higher temperatures. Please note that these two data sets have different spatial resolution and time series, which means the different sample numbers. However, it does not affect too much on the general climatology comparison. The new date set underestimates than observations except 1% quantile, which suggests that it may be warmer than observation in the cold
season. Only a 0.5 °C difference is found between the two data sets for 5% quantile. 1.3 °C, 1.9 °C and 1.8 °C is underestimated by the new data set for 95% quantile, 95% quantile and mean temperature, respectively. The average elevation of all 24 sites is 1862 m, which is 429 m higher than the average 1km grids height (1433 m). The altitude difference could explain partly for the bias in Table 3. It can be concluded that the new data set could generally capture the climatology of the entire CTM, especially for warmer temperatures.

**5 Discussion and conclusion**

Although the average temporal correlation (R = 0.986) between ERA-Interim and observations over the CTM is encouraging, an average RMSE of 3.75 °C suggests that a downscaling process of ERA-Interim is needed. There are many factors may lead to the errors such as assimilated observation errors, model background errors and operator errors in ECMWF system. However, previous studies have shown that the elevation difference between the ERA-Interim grid and the individual site is
the key factor for the errors in the high mountains such as the European Alps and the Tibetan Plateau (Gao et al., 2012, Gao et al., 2014a; Gao et al., 2017). Therefore, it is possible to reduce such errors as well as downscale the grid value to finer local scale via elevation-based methods.

Gao et al. (2012, 2017) claimed that the method based on ERA-Interim internal vertical lapse rates outperformed several conventional methods such as using fixed monthly lapse rates and observed lapse rates from meteorological stations in the
European Alps and the Tibetan Plateau. The performances were similar to ERA-Interim internal vertical lapse rates derived from different pressure levels in European Alps (Gao et al. 2012). Among these methods, using the lapse rate calculated





from the pressure levels, covering the highest and lowest meteorological stations, was proven more accurate in the Tibetan Plateau (Gao et al., 2017). The most prominent advantage is that this method is fully independent from the observed data. Therefore, it provides a possibility to extrapolate ERA-Interim temperature data for any other high mountain areas where no measurements exist.

Base on this hypothesis, the 0.25°×0.25°, 6-hourly ERA-Interim 2 m temperature data is downscaled to 1 km grid from 1979 to 2016 in the CTM, where is serious lack of long-term temperature observations. To our knowledge, the presented data is the first set (Version 1.0) with the high spatial-temporal resolution in such a long term series for this region. To evaluate the quality of the new data set, observations from 24 meteorological stations were used for comparison.  The average NSE enhances by 5% from 0.90 to 0.94 and an average 0.9 °C (24%) of RMSE is reduced from 3.75 to 2.85 °C. The average

PDF-based skill score increases from 0.86 to 0.91 after downscaling over all test sites. Except a few sites, the downscaling method has a good performance for most of sites, especially for the site located in the valley (No. 16 Baicheng). The PDF shape of new high-resolution grid data fits the observed PDF much better in comparison with the original ERA-Interim temperature. The validation indicates that the downscaling method is reliable and the RMSE and PDF discrepancy is reduced significantly for most of the stations.

The new data set captures the climatology of the entire CTM very well. It seizes the distribution characteristics like high temperature in the river valleys and basins and low temperature in the peaks. The bias is only 1.8 °C between the downscaled and the observations. Except the extreme cold temperature, the new data set has less 2 °C bias compared with observations. In general, the downscaled data set is appropriate for the climate impact assessment and has the potential to be used for hydrological and environmental modelling.

For sure, some issues about the quality of the data set as well as the validation should be addressed here. The main hypothesis is that the elevation plays the crucial role for temperature changes. It means that the temperature changes follow the lapse rate law in the vertical direction. However, in the horizontal direction, the micro-topographical features like aspect and slope of the mountain possibly affect the temperatures in a short time. Temperature can be significant different in the shady slope and sunny slope. The grid height of the data set is derived from SRTM DEM, which is re-sampled from 90 m to 1 km. As shown in Figure 1, the

highest altitude from the re-sampled SRTM is about 7100 m, which is lower around 300 m than the highest peak (Mount Tomur, 7443.8 m) in the CTM. Therefore, the DEM bias may lead to a systematic error compared with observations. The number of meteorological stations for validation here is limited to 24. These stations are mainly located in the valleys and basins. Thus, it is difficult to evaluate the credibility of data set in the high mountains where more glaciers covered. Other observational resources like remote sensing data are helpful for further validation. Due to the limitation of computational resource

and the accessibility of data source (only 6-hourly ERA-Interim temperatures is open access for public), the resolution of this data set is limited to 6-hourly and 1 km grid spacing. However, the current data set (~187 GB) is huge to process and store. The computational resource and the disk usage of the data set will increase exponentially when the spatial-temporal resolution becomes



higher. For such a huge amount of data, storage and extraction is not very convenient. Supercomputers and parallel computing are necessary in the future. Higher resolution, more validation and downscaling method improvement (Version 2.0) are the topics of on-going and future research.

## 6 Data availability

The data set presented in this article is published in Network Common Data Form (NetCDF) at doi: 10.1594/PANGAEA.887700. The coverage of data set is 41.1814-45.9945 °N, 77.3484-96.9989 °E. The grid point is derived from SRTM DEM, which is re-sampled from 90 m to 1 km. The total number of grid points is 818126. The time step is 6 hourly at 00, 06, 12, and 18 UTC. The time series is from 1979 to 2016. In order to reduce the size of each file, the grid points (818126) are divided into 41 groups. Thus, each nc file contains around 20000 grid points, which according to grid

point ID (see grid_points.nc). The time series also is divided by five years. The nc file name specifically show the data information. For example, t2m_1979_1984_1_20000.nc means t2m_'beginning year'_'ending year'_'beginning grid point'_'ending grid point'.nc. The total number of nc files is 288. The disk usage of the data set is about 187 GB. The users can access the data via DOI link to the PANGAEA webpage (View dataset as HTML under the Download Data item) in the following steps: 1) download grid_points.nc file, and then select the grid points for the target study area according to the

coordinates. For example, a study area covers the grid points ID from 200 to 1000, and 14000 to 25000. The time series is supposed to be 1985 to 1989; 2) download the data file t2m_1985_1989_1_20000.nc, and t2m_1985_1989_20001_40000.nc; 3) extract the temperature data according to the grid point IDs; 4) analysis data and plot figures.

**Competing interests.** The authors declare that they have no conflict of interest.

**Acknowledgements.** This study was supported by the National Natural Science Foundation of China (grant number
41501106). Dr. Jianhui Wei was supported financially by the German Research Foundation through funding of the AccHydro project (DFG-grant KU 2090/11-1). ERA-Interim data was supported by the ECMWF (https://www.ecmwf.int/en/forecasts/datasets). The meteorological data have been provided by China Meteorological Data Sharing Service System of National Meteorological Information Center (CMA-CMDC, http://data.cma.cn/). The 7 out of 24 sites for global exchange are provided by CMDC
(http://data.cma.cn/en/?r=data/detail&dataCode=SURF_CLI_CHN_MUL_DAY_CES_V3.0).The authors also thank the PANGAEA, Data Publisher for Earth & Environmental Science to publish our data set as open access to public users (https://www.pangaea.de/).





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



Table 1. Test sites information (ERA_height is the ERA-Interim grid height)

| ID | Name | Latitude (°) | Longitude (°) | Elevation (m) | ERA_height (m) | Initial time |
|----|------|--------------|---------------|---------------|----------------|--------------|
| 1 | Wenquan | 44.98 | 81.07 | 1133 | 1866 | 1979-01-01 |
|   |         | 44.97 | 81.02 | 1355 | 1866 | 1981-01-01 |
| 2 | Jinghe | 44.62 | 82.90 | 320 | 1133 | 1979-01-01 |
|   |        | 44.60 | 82.90 | 319 | 1133 | 2006-03-01 |
| 3 | Wusu | 44.43 | 84.67 | 479 | 963 | 1979-01-01 |
| 4 | Caijiahu | 44.20 | 87.53 | 441 | 913 | 1979-01-01 |
| 5 | Qitai | 44.02 | 89.57 | 794 | 1258 | 1979-01-01 |
| 6 | Yining | 43.95 | 81.33 | 663 | 1325 | 1979-01-01 |
| 7 | Zhaosu | 43.15 | 81.13 | 1851 | 1963 | 1979-01-01 |
| 8 | Urumqi | 43.78 | 87.62 | 918 | 1492 | 1979-01-01 |
|   |        | 43.78 | 87.65 | 935 | 1458 | 2000-01-01 |
| 9 | Baluntai | 42.67 | 86.33 | 1753 | 2616 | 1979-01-01 |
|   |          | 42.73 | 86.30 | 1737 | 2616 | 1994-12-01 |
| 10 | Dabancheng | 43.35 | 88.32 | 1104 | 1491 | 1979-01-01 |
| 11 | Qijiaojing | 43.48 | 91.63 | 873 | 1232 | 1979-01-01 |
|    |            | 43.22 | 91.73 | 790 | 1077 | 1999-01-01 |
| 12 | Kumishen | 42.23 | 88.22 | 922 | 1305 | 1979-01-01 |
| 13 | Bayinbrook | 43.03 | 84.15 | 2458 | 2841 | 1979-01-01 |
| 14 | Yanqi | 42.08 | 86.57 | 1055 | 1618 | 1979-01-01 |
| 15 | Turfan | 42.93 | 89.20 | 35 | 1115 | 1979-01-01 |
| 16 | Baicheng | 41.78 | 81.90 | 1229 | 1730 | 1979-01-01 |
| 17 | Luntai | 41.78 | 84.25 | 976 | 1338 | 1979-01-01 |
|    |        | 41.82 | 84.27 | 982 | 1338 | 2011-01-01 |
| 18 | Kuche | 41.72 | 82.95 | 1099 | 1460 | 1979-01-01 |
|    |       | 41.72 | 83.07 | 1082 | 1460 | 1993-01-01 |
| 19 | Kuerle | 41.75 | 86.13 | 932 | 1245 | 1979-01-01 |
| 20 | Balitang | 43.73 | 93.07 | 1638 | 1549 | 1979-01-01 |
|    |          | 43.60 | 93.00 | 1165 | 1482 | 1985-01-01 |
|    |          | 43.60 | 93.05 | 1677 | 1482 | 2003-07-01 |
| 21 | Naomaohu | 43.77 | 95.13 | 479 | 1066 | 2013-12-31 |
| 22 | Yiwu | 43.27 | 94.70 | 1729 | 1494 | 2013-12-31 |
| 23 | Hami | 42.82 | 93.52 | 737 | 1208 | 2013-12-31 |
| 24 | Hongliuhe | 41.53 | 94.67 | 1170 | 1450 | 2002-06-30 |
|    |           | 41.53 | 94.67 | 1568 | 1450 | 2002-07-01 |



Table 2. Comparison of original and downscaled ERA-Interim temperature with daily observations. The NSE, PDF-based skill score as well as RMSE in °C are listed.

| ID | NSE | | RMSE | | PDF-based skill score | |
|---|---|---|---|---|---|---|
| | original | downscaled | original | downscaled | original | downscaled |
| 1 | 0.92 | 0.94 | 3.61 | 3.07 | 0.82 | 0.90 |
| 2 | 0.93 | 0.92 | 3.89 | 4.32 | 0.83 | 0.89 |
| 3 | 0.95 | 0.96 | 3.47 | 2.95 | 0.84 | 0.92 |
| 4 | 0.93 | 0.92 | 4.23 | 4.75 | 0.81 | 0.87 |
| 5 | 0.97 | 0.96 | 2.81 | 3.01 | 0.88 | 0.89 |
| 6 | 0.89 | 0.96 | 3.86 | 2.25 | 0.86 | 0.94 |
| 7 | 0.94 | 0.95 | 2.58 | 2.32 | 0.89 | 0.91 |
| 8 | 0.89 | 0.96 | 4.57 | 2.61 | 0.79 | 0.87 |
| 9 | 0.46 | 0.82 | 7.76 | 4.47 | 0.67 | 0.76 |
| 10 | 0.96 | 0.98 | 2.35 | 1.83 | 0.92 | 0.93 |
| 11 | 0.93 | 0.96 | 3.68 | 2.82 | 0.88 | 0.92 |
| 12 | 0.94 | 0.97 | 3.33 | 2.32 | 0.88 | 0.95 |
| 13 | 0.78 | 0.69 | 6.65 | 7.80 | 0.71 | 0.75 |
| 14 | 0.92 | 0.97 | 3.39 | 2.27 | 0.85 | 0.93 |
| 15 | 0.71 | 0.94 | 7.69 | 3.45 | 0.76 | 0.92 |
| 16 | 0.95 | 0.93 | 2.61 | 3.14 | 0.92 | 0.86 |
| 17 | 0.96 | 0.98 | 2.53 | 1.53 | 0.93 | 0.95 |
| 18 | 0.93 | 0.98 | 3.17 | 1.63 | 0.90 | 0.95 |
| 19 | 0.92 | 0.98 | 3.39 | 1.78 | 0.88 | 0.97 |
| 20 | 0.94 | 0.97 | 3.19 | 2.32 | 0.89 | 0.93 |
| 21 | 0.92 | 0.98 | 4.19 | 2.02 | 0.86 | 0.98 |
| 22 | 0.95 | 0.97 | 2.49 | 2.03 | 0.89 | 0.93 |
| 23 | 0.96 | 0.98 | 2.56 | 2.00 | 0.93 | 0.94 |
| 24 | 0.97 | 0.98 | 2.05 | 1.60 | 0.94 | 0.96 |
| Average | 0.90 | 0.94 | 3.75 | 2.85 | 0.86 | 0.91 |



Table 3. Climatology of the entire CTM based on downscaled ERA-Interim temperature (1km, 6-hourly, 1979-2016) and daily observations (daily, 1979-2013, °C).

|  | 1% quantile | 5% quantile | 95% quantile | 99% quantile | Mean |
|---|---|---|---|---|---|
| downscaled ERA-Interim | -20.6 | -15.5 | 25.9 | 29.5 | 6.0 |
| observation | -23.2 | -15.0 | 27.2 | 31.4 | 7.8 |



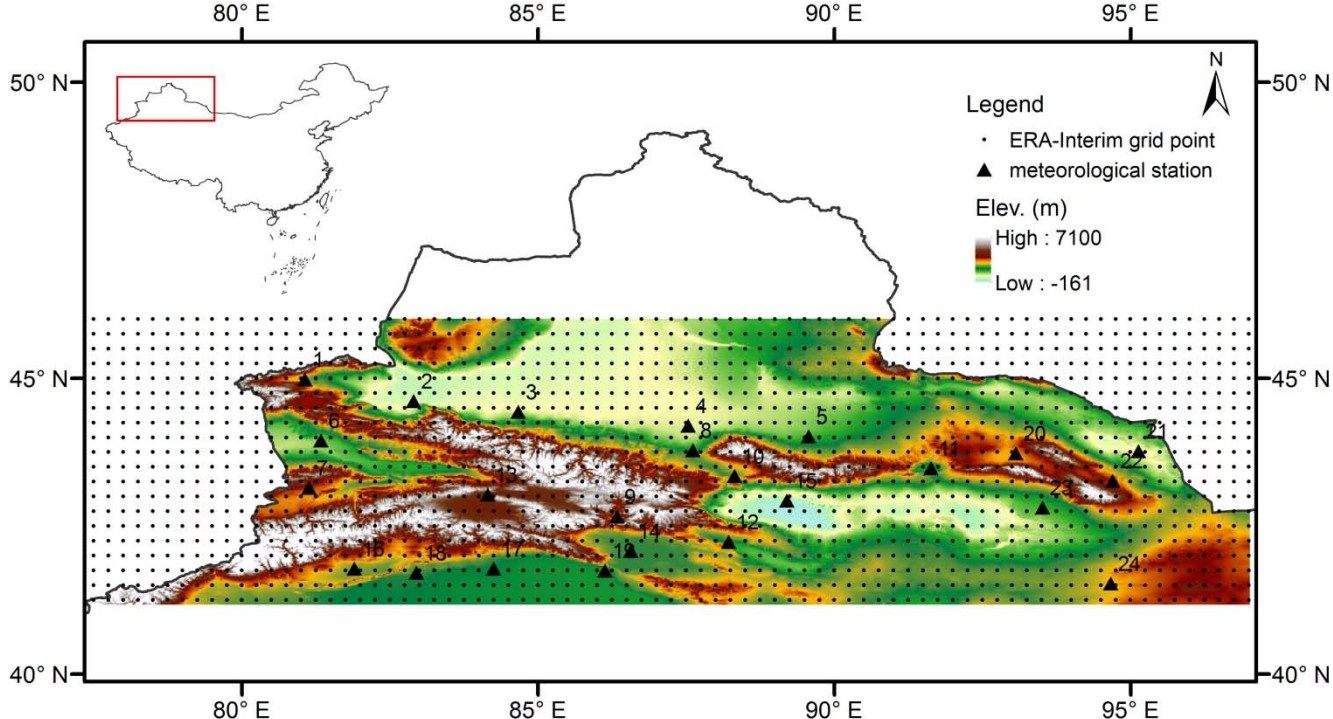

**Figure 1:** **Location of the 24 meteorological stations (triangle) and ERA-Interim 0.25°×0.25° grid points (dot). The elevation ranges from -161 m to 7100 m a.s.l., with a DEM resolution of 1km.**





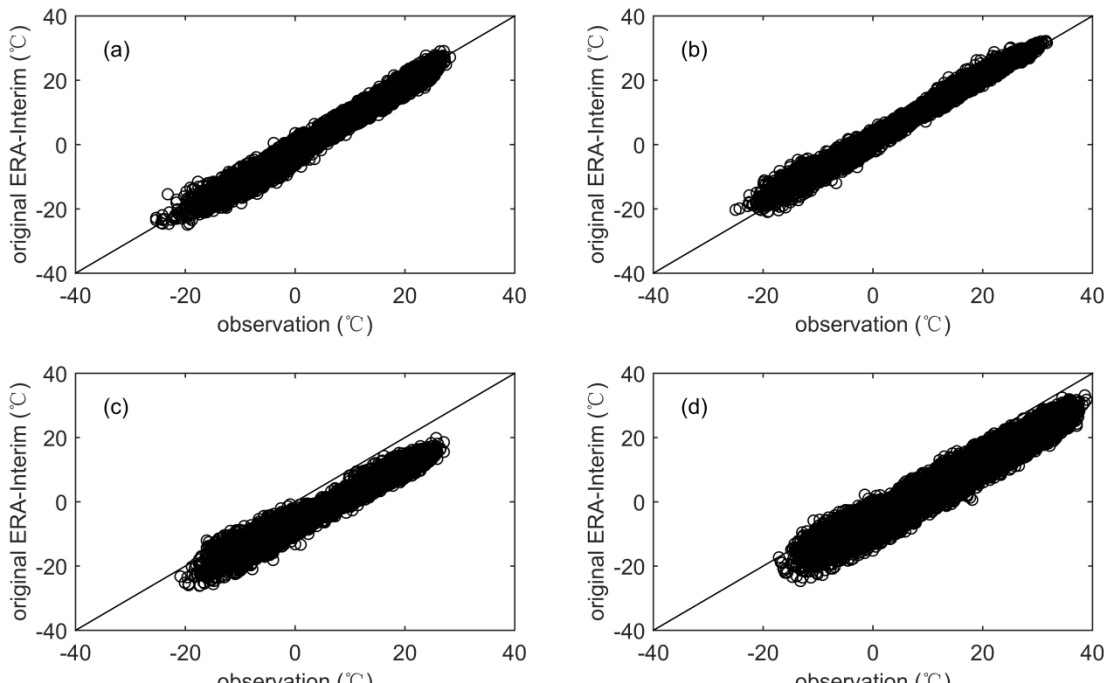

**Figure 2: Scatter plots of observation and original ERA-Interim temperature data for (a) station No. 10, (b) station No. 24, (c) station No. 9 and (d) station No. 15. The corresponding RMSE can be found in Table 2.**




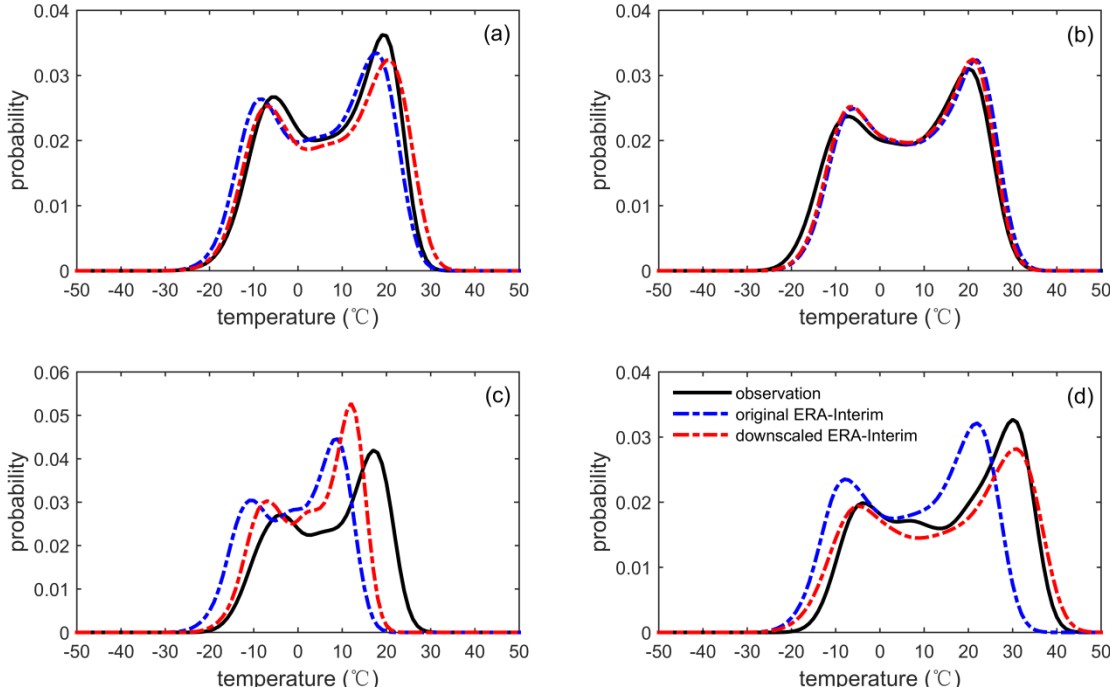

**Figure 3: Probability density functions for observation, original ERA-Interim and downscaled ERA-Interim for (a) station No. 10, (b) station No. 24, (c) station No. 9 and (d) station No. 15. The corresponding PDF-based skill score can be found in Table 2.**




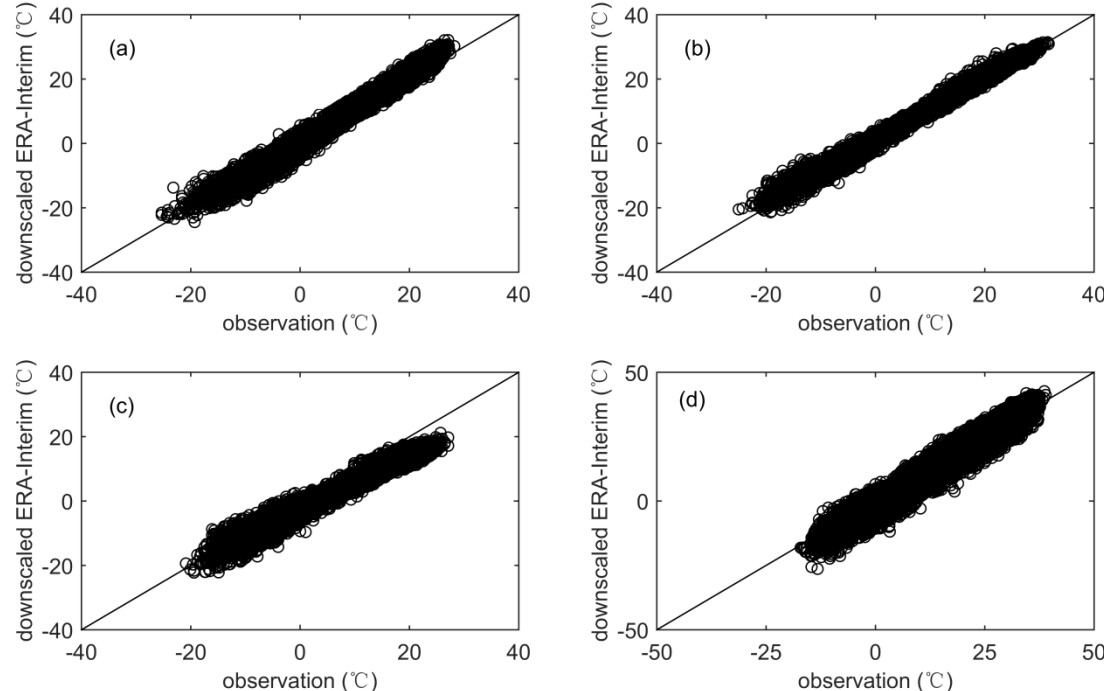

**Figure 4: Scatter plots of observation and downscaled ERA-Interim temperature data for (a) station No. 10, (b) station No. 24, (c) station No. 9 and (d) station No. 15. The corresponding RMSE can be found in Table 2.**





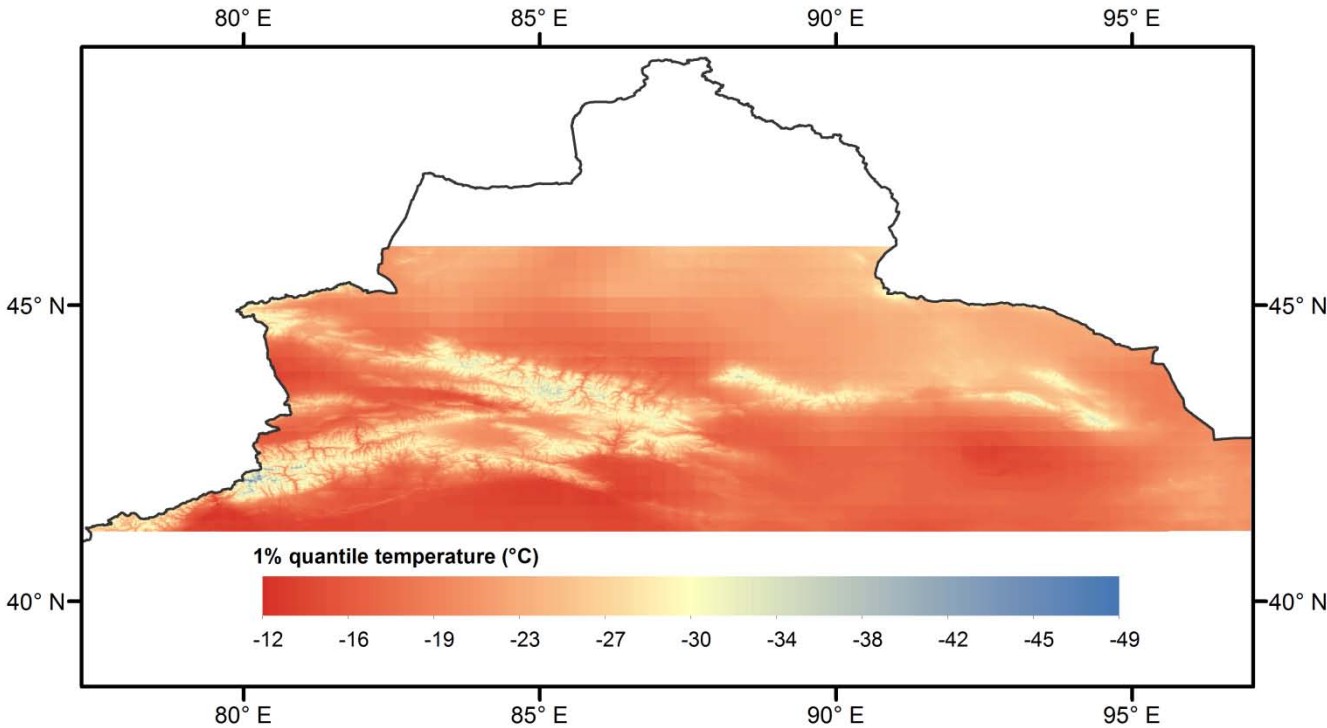

**Figure 5: 1% quantile temperature of the entire CTM based on the high-resolution data set.**



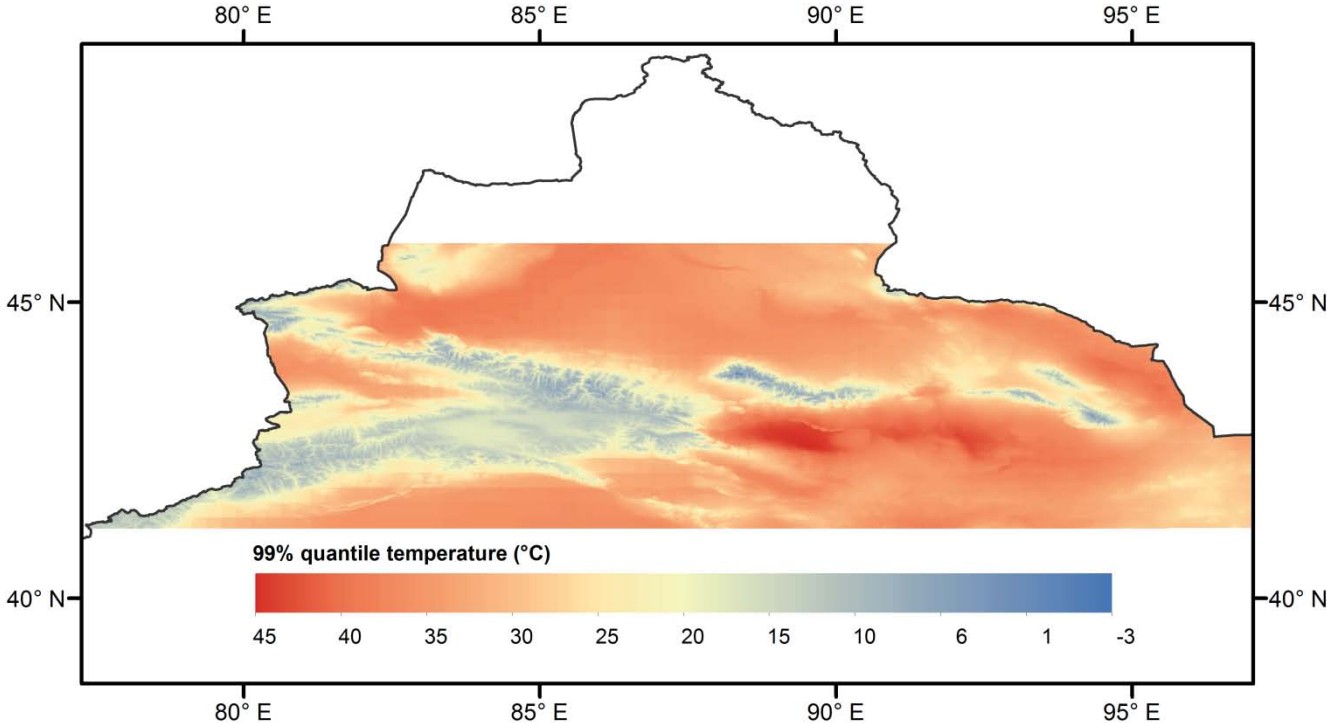

**Figure 6: 99% quantile temperature of the entire CTM based on the high-resolution data set.**



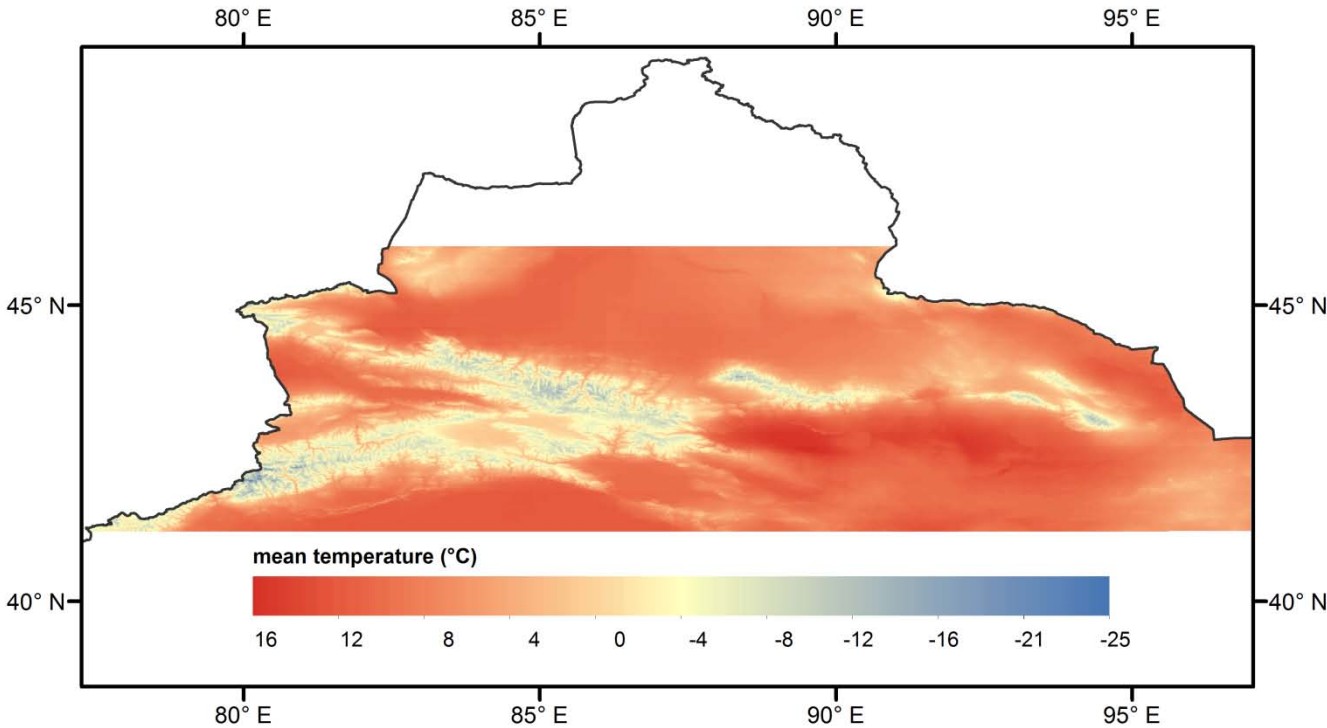

**Figure 7: Mean temperature of the entire CTM based on the high-resolution data set.**