# Peer review of "A high-resolution air temperature data set for the Chinese Tianshan Mountains in 1979-2016"

_Earth System Science Data, 2018_

## Referee Comment (RC1) · L. Gerlitz (Referee) · 9 Jul 2018

The presented manuscript introduces a newly established data set of elevation corrected 6-hourly near surface temperatures at a resolution of 1km for the Tianshan mountains. Temperature lapse rates are derived from free atmospheric ERA-Interim data at various pressure levels and are interpolated to high spatial resolution. The ERA-Interim internal lapse rates are subsequently used to correct the near surface temperatures under consideration of a high resolution DEM. The data set is evaluated against observations (24 meteorological stations are considered) and the general characteristics of the spatial temperature distribution over the target domain are presented. In general the target of the study is timely, since high resolution climate data represent an important input for many climate impact modelling applications. However, in my

opinion the evaluation of the data set needs to be improved in order to better communicate its limitations to potential users. Further I would suggest to better investigate the major characteristics of the temperature distribution over the Tienshan mountains and to propose some potential applications.

In the following I will summarize my major concerns without going into detail:

1. Terminology and Language: The applied methods are presented as a downscaling technique. In the introduction the authors state that important local-scale processes, such as cold-air pooling or snow-melt related processes are not represented by reanalysis products due to their limited spatial resolution. However, the suggested elevation correction technique does not consider such processes and thus should not be termed as a downscaling technique. I suggest to use "elevation adjustment" throughout the manuscript.

In general, the language of the manuscript is somehow unprecise or misleading at some points. I suggest to include a native speaker.

2. Data and methods: The introduction of the utilized data sets is very short and some of the applied techniques are not fully clarified. - Which levels are used for the elevation adjustment of a specific pixel? There is some information given on page 5, l. 15, but unfortunately I cannot follow. Maybe it would be helpful to provide a brief example. In general I suggest to describe the elevation correction technique in greater detail! - Which ERA-Interim data are used? I suppose the authors make use of the fully assimilated data set, however p.4,l22 discusses the 10 days forecast. Please clarify. - For the Evaluation 24 records are used. Are these stations independent of the reanalysis, i.e. they are not used for the assimilation? If they are part of the assimilation procedure, the skill of ERA-Interim might be overestimated.

3. Evaluation: - The Evaluation of the data set is done against 24 meteorlogical stations. Therefor the modelled temperature is derived by averaging (?) the 3*3 gridcells surrounding each climate station. This approach unfortunately leads to a systematic

bias of the modelled temperature data, since the station elevation does not coincide with the mean elevation of the considered grid cell. Further, the spatial averaging generates a smoothened temperature field, i.e. the data set is actually not evaluated at a 1km resolution, but at 3km. I would highly suggest to improve the evaluation methodology. In order to completely overcome the systematic bias, the lapse rates could be used to adjust the temperature directly to the elevation of the meteorological station (without condsidering the DEM). Therefor the ERA-Interim internal lapse rate of the corresponding pixel could be employed. Most likely this will lead to a better skill. A temperature bias of 3°C is still a lot and is probably due to the elevation induced systematic bias. - The evaluation correction is conducted for different periods (p. 6,l15). I would suggest compare the period from 1979 to 2013 only. If the quality of the data set is good enough, the data set can still be extended for the remaining years. - The data set includes 6-hourly values, however the evaluation is only conducted for aggregated measures, such as mean, max and min. It is very likely, that the quality of the data set varies in different seasons and different times of the day. E.g. cold air pooling during winter nights might lead to a strong warm bias of the data set, strong diurnal heating during the day may have opposite effects (see e.g. (Gerlitz 2014)). I suggest to test the quality of the data set for different seasons and times of the day independently, in order to communicate the limitations of such an approach to potential users. - Evaluation of lapse rates: Usually the lapse rates in high mountain regions have typical diurnal and seasonal cycles. However, the free air lapse rates might not correspond with lapse rates at the surface. I would like to see a brief evaluation of the lapse rates which are used for the elevation adjustment. Do they correspond with observations? Do they have any spatial variations? The authors e.g. state that the data set slightly improves ERA-Interim data for some locations, particularly for higher temperatures (p8,l6). Does that mean, that winter lapse rates are not well simulated by ERA-Interim? - The section on the evaluation measures of specific stations is lengthy and difficult to follow. The authors mention the number and the performance measures for each station and mention that the approach does not work well for all sites (p8,l23). Would it be possible, to in-
terpret the differences of the model skill with regard to potential local scale processes, that are not captured by your approach? I could imagine that stations in deep valleys react differently compared with stations located at higher elevations. A comprehensive interpretation of the data quality would inform potential users about the stengthes and limitations of the data set.

4. Application of the data set The authors show very general characteristics of the data set, such as mean, minimum and maximum temperatures, in section 4.3. Most applications, which are mentioned in the introduction, however require both, high resolution temperature and precipitation data. I feel that the potential of such a data set should be better illustrated by showing its unique features. Does the high resolution data set e.g. reproduces elevation depending warming in the Tienshan mountains? (see e.g. (Gerlitz et al. 2014)). Are spatial and seasonal variations of the dirurnal temperature range well captured (Sun et al. 2018; Shekhar et al. 2018)? Such potential applications could be included without much effort and will certainly illustrate the potential of the data set, which stands out due to its spatial AND temporal resolution.

5. Data Availability The structure of the data set seems to be a bit unintuitive to me. Wouldn't it be an option to provide the NCDF files for each year and for the entire domain? This would simplify the usaga of the data set, particularly for users who download the data set via batch scripts.

Gerlitz, L., 2014: Using fuzzified regression trees for statistical downscaling and regionalization of near surface temperatures in complex terrain. Theor Appl Climatol, 122, 337–352, doi:10.1007/s00704-014-1285-x. Gerlitz, L, O. Conrad, A. Thomas, and J. Böhner, 2014: Warming patterns over the Tibetan Plateau and adjacent lowlands derived from elevation- and bias‑corrected ERA-Interim data. Climate Research, 58, 235–246, doi:10.3354/cr01193. Shekhar, M. S., U. Devi, S. K. Dash, G. P. Singh, and A. Singh, 2018: Variability of Diurnal Temperature Range During Winter Over Western Himalaya: Range- and Altitude-Wise Study. Pure Appl. Geophys., 1–13, doi:10.1007/s00024-018-1845-6. Sun, X., and Coauthors, 2018: Global diurnal

temperature range (DTR) changes since 1901. Clim Dyn, 1–14, doi:10.1007/s00382-018-4329-6.

---

## Referee Comment (RC2) · Anonymous Referee #2 · 20 Jul 2018

The authors present a high-resolution (1km, 6h) air temperature data set for the Chinese Tianshan Mountains from 1979 to 2016 based on a downscaling method. This topic is quite interesting and the data set would be useful for the potential end-users who focus on the alpine climate and cryosphere issues in the Tianshan Mountains. In general, this paper is well-written for most parts. However, a major revision is needed before it is published in ESSD. General Comments: 1. The mentioned downscaling method in the paper has been validated in the Alps Mountains and the Tibet Plateau. However, the relative references (Gao et al., 2012, 2017) named this method as "elevation correction" rather than "downscaling". What is the difference between these two methods or terms? For me, they are the same. Thus, which one is more appropriate? 2. 24 meteorological stations are not enough for validation for such

a large area (more than 80000 points). Is there any other data resources could be used for further validation? 3. The authors pointed out that about 24% of RMSE was reduced by the downscaling method compared to the original ERA-Interim. Is it good enough? How to evaluate the data set (or any reference/standard) is good enough for end-users? 4. For my understanding, the downscaling method is mainly based on the elevation (DEM). Is it possible to get higher resolution data set if we use the 100 m DEM? The ERA-Interim product provides 3-hourly forecast data. Thus, is it possible to obtain 3-hourly data set for whole Tianshan Mountains? 5. How to evaluate the lapse rate is correct or appropriate for the downscaling? The lapse rate varies significant in different topographical situations and time period. 6. Precipitation is another basic and important variable for climate and environmental models. Can you produce any high resolution precipitation data set using some similar methods for this region? Specific comments: 1. Although the authors listed many references about the downscaling method, I believe it is necessary to clarify the method specific for the readers who are not familiar with downscaling method. 2. The downscaling method is more appropriate named "elevation correction". Since only elevation is involved. The conventional circulation variables such as wind, sea level pressure, humidity are not considered in the downscaling method. 3. The data set is not friendly to download and use. The data set is divided into so many sub-files. Is it possible to find a more easy way for users? 4. The downscaled data at some sites are worse than the original ERA-Interim data. Why? The authors should discuss this issue. It is very important because only 24 sites are available for validation. 5. If someone plans to run a hydrology model in a small catchment in the Tianshan Mountains. How to adjust the data set points to match the model grids? 6. I found that the data amount is around 187G. How to process such large data set? What is software or platform to process it? Maybe, the authors could provide some codes for data processing. 7. I am not sure the data set could capture the temperature changes in the micro-topography since the original data is 0.25 degree. The slope and aspect of mountains also affect the temperature significant, especially in the night. 8. Can other temperature downscaling methods be used for the

high-resolution data set? And why? 9. Some expression and description of language is not clear. A native speaker would be helpful for the improvement of readability for the whole context.

---

## Referee Comment (RC3) · Anonymous Referee #3 · 26 Jul 2018

This is an interesting piece of research that are focusing on downscaling temperature data for one mountainous region in China. The study is clearly written and the data/method is solid. I recommend a "minor revision" for this paper. Below is my comments and questions. 1. Needs a short introduction of the mechanism of the method mentioned in line 31, page 2. 2. The labels for stations in figure 1 are not very clear, please use another color. 3. In figure 2 and 4, why Nos.10, 24, 9 and 15 stations are chosen for comparison? Are they the most representative ones? 4. Although the author mentioned that the temporal resolution of the new dataset is 6 hours, non of the analysis is focused on the temporal variations of the temperature. Needs at least some analysis to address the temporal accuracy of the new dataset. 5. Page 9, line 23, delete "there are". 6. How the comparison error statistics are sensitive to the "down-

scaling resolution" you have chosen? Why the final product is set to 1 km grids? 7. Have you considered the diurnal changes of the situation in mountainous areas. Since the data you have corrected is at 6 hour resolution. In mountainous area, you may expect temperature inversions during night time.

---

## Short Comment (SC1) · 17 Aug 2018

Producing the high-resolution air temperature dataset in the mountain area is useful for regional climate or hydrological studies. The general target of this study is important. I have some suggestions and comments in below:

1. The method and the used lapse-rates should be more detail described in section 3.3.

2. In table 1, is the elevation of the sites as same as the height from the 1 km SRTM DEM grid? it seems not the same (from P9, I18 ., I recommend to list the sites height in 1km DEM.

3. Addition to comment 2. As the height in DEM grid and site point is different, in P6

I13, why do you use averaged 9 grid points to evaluate the downscaled results. In my opinion, since each point in 1 km grid is downscaled according to its DEM height and ERA height, the nest grid point or the nest height grid of 9 point should be used for comparison.

4. The lapse rate varies in the different topographical situation and different timestamp like during the nighttime (Li et. al. 2014). The authors should discuss more about the method and results on the diurnal scale.

5. The authors produce the data from ∼25 km to 1 km resolution. The total grid points are 818126, however, only 24 observation stations are used in validation. And in Figure 2-4, the authors only present comparison results at 4 stations, which probably have the best results. In addition, the authors only validate this 6-hourly dataset at a daily scale. To my point, the validation is somehow insufficient. It is not enough to conclude the reliability of this dataset, at least in the current level of discussion of the manuscript. I recommend the authors present some comparison analysis with ERA-Interim on the diurnal scale, and have more validation results with the station observations. Although very limited stations exist in this area, I know the diurnal max. and min. temperatures are provided in CMA station datasets, and these could be used in robust validation.

6. As the dataset provide temperature in 38 years, the authors should show some validation in annual and seasonal scale for this long period. In the spatial scale, I recommend to validate the dataset in each sub regions based on hydrology basin or climate zone or different elevation ranges. Figure 5 and figure 6 can not give much information of any performance skill of this dataset.

7. The dataset is very not friendly to use to me. As NetCDF format, I recommend to provide each file for the whole area at each timestamp (or each day, each month, each year) like most Grid datasets did (APHRODITE, TRMM, China meteorological forcing data from CAS. et. al. ). It will be much easier for regional climate or hydrology studies. Or save as the GRIB format like the reanalysis dataset.

Reference:

Li, Y., Z. Zeng, L. Zhao, and S. Piao ( 2015), Spatial patterns of climatological temperature lapse rate in mainland China: A multi–time scale investigation, J. Geophys. Res. Atmos., 120, 2661–2675, doi:10.1002/2014JD022978.

---

## Author Comment (AC1) · 16 Sep 2018

**Response to Reviewer 1**

We would like to thank Dr. Lars Gerlitz for reviewing our manuscript. These comments are really helpful for improving the manuscript. In the following, we address all comments point-by-point according to reviewers' comments. All revisions are highlighted in the context.

The presented manuscript introduces a newly established data set of elevation corrected 6-hourly near surface temperatures at a resolution of 1km for the Tianshan mountains. Temperature lapse rates are derived from free atmospheric ERA-Interim data at various pressure levels and are interpolated to high spatial resolution. The ERA-Interim internal lapse rates are subsequently used to correct the near surface temperatures under consideration of a high resolution DEM. The data set is evaluated against observations (24 meteorological stations are considered) and the general characteristics of the spatial temperature distribution over the target domain are presented. In general the target of the study is timely, since high resolution climate data represent an important input for many climate impact modelling applications. However, in my opinion the evaluation of the data set needs to be improved in order to better communicate its limitations to potential users. Further I would suggest to better investigate the major characteristics of the temperature distribution over the Tienshan mountains and to propose some potential applications.

-Answer: Thanks a lot for the comments. It is true that high resolution data set is extreme needed for the TianShan Mountains, not only temperature but also precipitation and other variables. Dr. Gerlitz pointed out a very important issue on the evaluation and further application. We tried to answer the questions and improve the manuscript. The details are presented in the following context.

In the following I will summarize my major concerns without going into detail:

1. Terminology and Language: The applied methods are presented as a downscaling technique. In the introduction the authors state that important local-scale processes, such as cold-air pooling or snow-melt related processes are not represented by reanalysis products due to their limited spatial resolution. However, the suggested elevation correction technique does not consider such processes and thus should not be termed as a downscaling technique. I suggest to use "elevation adjustment" throughout the manuscript. In general, the language of the manuscript is somehow unprecise or misleading at some points. I suggest including a native speaker.

-Answer: Thanks a lot for the comments. Yes, we corrected the ERA-Interim data using a lapse rate scheme without considering the local-scale processes. Although an elevation correction is a form of downscaling, we agree that the term "elevation correction" is more intuitive than "downscaling" for readers' better understanding. We revised this term in the revision version. Meanwhile, we asked the Elsevier publishing group (https://webshop.elsevier.com/) for help to correct our terminology and language problems.

2. Data and methods: The introduction of the utilized data sets is very short and some of the applied techniques are not fully clarified. - Which levels are used for the elevation adjustment of a specific pixel? There is some information given on page 5, l. 15, but unfortunately I cannot follow. Maybe it would be helpful to provide a brief example. In general I suggest to describe the elevation correction technique in greater detail! - Which ERA-Interim data are used? I suppose the authors make use of the fully assimilated data set, however p.4,l22 discusses the 10 days forecast. Please clarify. - For the Evaluation 24 records are used. Are these stations independent of the reanalysis, i.e. they are not used for the assimilation? If they are part of the assimilation procedure, the skill of ERA-Interim might be overestimated.

--Answer: Thanks a lot for pointing this out. We agree that the method should

be presented in more detail. We added more information on the correction method, especially an example on the internal lapse rate scheme (**P6 L2-11**).

Dr. Gerlitz also raised a very important issue that if some individual sites are assimilated by ECMWF Integrated Forecast System (IFS), the ERA-interim predictions are not fully independent from the observed data which are subsequently used for calibration and validation. We investigated the ECMWF assimilation records and found that 9 of 24 sites were possible assimilated by IFS. Table 1 shows the sites details. The long-term temperature records (1979-2011) from Nos. 6, 8, 18 and 23 were assimilated. Only short-term observations (less than 15 years) from other 5 sites were assimilated. According to the information of the ECMWF, it can be assumed that although 9 of 24 sites were possible assimilated, other 15 sites were not used by ERA-Interim and therefore represent fully independent data set. Furthermore, compared with the assimilated short-term observations, we tested much longer time series. Thus, we believe the skill of ERA-Interim is not affected (**P5 L16-24**).

Table 1 Assimilated sites in ERA-Interim.

| ID | Name | WMO id | starting date | ending date |
|---:|------|-------:|---------------|-------------|
| 2 | Jinghe | 51334 | 1979-06-21 | 1993-01-21 |
| 5 | Qitai | 51379 | 1979-06-03 | 1985-05-20 |
| 6 | Yining | 51431 | 1978-12-31 | 2011-12-31 |
| 8 | Urumqi | 51463 | 1978-12-31 | 2011-12-31 |
| 11 | Qijiaojing | 51495 | 1979-04-07 | 1993-04-24 |
| 15 | Turfan | 51573 | 1981-06-30 | 1984-08-08 |
| 18 | Kuche | 51644 | 1978-12-31 | 2011-12-31 |
| 19 | Kuerle | 51656 | 1979-01-03 | 1994-12-30 |
| 23 | Hami | 52203 | 1978-12-31 | 2011-12-31 |

3. Evaluation: The Evaluation of the data set is done against 24 meteorological stations. There for the modeled temperature is derived by averaging (?) the 3*3 grid cells surrounding each climate station. This approach unfortunately leads to a systematic bias of the modelled temperature data, since the station

elevation does not coincide with the mean elevation of the considered grid cell. Further, the spatial averaging generates a smoothened temperature field, i.e. the data set is actually not evaluated at a 1km resolution, but at 3km. I would highly suggest to improve the evaluation methodology. In order to completely overcome the systematic bias, the lapse rates could be used to adjust the temperature directly to the elevation of the meteorological station (without condsidering the DEM). Therefore the ERA-Interim internal lapse rate of the corresponding pixel could be employed. Most likely this will lead to a better skill. A temperature bias of 3 degree is still a lot and is probably due to the elevation induced systematic bias. - The evaluation correction is conducted for different periods (p. 6, l15). I would suggest compare the period from 1979 to 2013 only. If the quality of the data set is good enough, the data set can still be extended for the remaining years. - The data set includes 6-hourly values; however the evaluation is only conducted for aggregated measures, such as mean, max and min. It is very likely, that the quality of the data set varies in different seasons and different times of the day. E.g. cold air pooling during winter nights might lead to a strong warm bias of the data set, strong diurnal heating during the day may have opposite effects (see e.g. (Gerlitz 2014)). I suggest to test the quality of the data set for different seasons and times of the day independently, in order to communicate the limitations of such an approach to potential users. - Evaluation of lapse rates: Usually the lapse rates in high mountain regions have typical diurnal and seasonal cycles. However, the free air lapse rates might not correspond with lapse rates at the surface. I would like to see a brief evaluation of the lapse rates which are used for the elevation adjustment. Do they correspond with observations? Do they have any spatial variations? The authors e.g. state that the data set slightly improves ERA-Interim data for some locations, particularly for higher temperatures (p8, l6). Does that mean, that winter lapse rates are not well simulated by ERA-Interim? - The section on the evaluation measures of specific stations is lengthy and difficult to follow. The authors mention the number and the

performance measures for each station and mention that the approach does not work well for all sites (p8,l23). Would it be possible, to interpret the differences of the model skill with regard to potential local scale processes, those are not captured by your approach? I could imagine that stations in deep valleys react differently compared with stations located at higher elevations. A comprehensive interpretation of the data quality would inform potential users about the strengths and limitations of the data set.

--Answer: Thanks a lot for the comments. Dr. Gerlitz pointed out a very important issue. It is true that the modeled temperature is averaged by the 3*3 grid cells surrounding each station. The systematic bias is negligible since the elevation differences are very tiny (smaller than 2m) among the 9 grids at 1km *1km grid resolution. When the authors evaluated the ERA-Interim temperature over the Tibetan Plateau (Gao et al., 2014), one reviewer suggested to select 3*3 grids with the station located in the center grid. He/she claimed that this way can evaluate the ability of ERA-Interim on different topographies. Thus, in this study we took this suggestion (**P7 L12-15**). This approach may lead to a systematic bias since the station elevation does not coincide with the mean elevation of the considered grid cells perfectly. However, the elevation differences between averaged 9 grids and station elevations are quite small with an average of -8 m (Table 2). Except the No. 9, the rest stations have less than 50 m elevation differences. From this point view, the systematic bias is very small. And the DEM generally matches the station elevations (**P7 L16-21**).

Table 2 Elevation of averaged 9 grids and the elevation differences with station elevation (m).

| ID | averaged 9 grids elevation | elevation Difference |
|----|-----|-----|
| 1 | 1305 | 50 |
| 2 | 306 | 14 |
| 3 | 477 | 2 |

| | | |
|---|---|---|
| 4 | 467 | -26 |
| 5 | 764 | 30 |
| 6 | 672 | -9 |
| 7 | 1885 | -34 |
| 8 | 893 | 25 |
| 9 | 2004 | -251 |
| 10 | 1101 | 3 |
| 11 | 868 | 5 |
| 12 | 940 | -18 |
| 13 | 2462 | -4 |
| 14 | 1057 | -2 |
| 15 | 11 | 24 |
| 16 | 1221 | 8 |
| 17 | 978 | -2 |
| 18 | 1066 | 33 |
| 19 | 937 | -5 |
| 20 | 1635 | 3 |
| 21 | 433 | 46 |
| 22 | 1814 | -85 |
| 23 | 758 | -21 |
| 24 | 1548 | 20 |
| Average | | **-8** |

Dr. Gerlitz suggested adjusting the temperature directly to the elevation of the meteorological station. For sure, we can correct the temperature for individual sites (selected the closest grid), just like the studies we have done in the Alps and the Tibet Plateau (Gao et al., 2012, 2017). Table 3 shows the RMSEs of the original and corrected ERA-Interim temperature using 9 grids as well as directly using station elevations. The RMSE differences between two approaches are small for all most of the sites (averaged RMSE only 0.05 °C). It is true that the bias is reduced more significant such as Nos. 20 and 24 using the station elevation directly. However, our goal is to produce continuous spatial-temporal data sets based on DEM, which could be easy applied for such hydrology and regional climate models. The surface sites are only used for validate the quality of data set.

Table 3 RMSEs of the original and corrected ERA-Interim temperature using 9

grids as well as directly with station elevations.

| ID | original ERA-Interim | corrected based on 9 DEM grids | corrected directly based on station elevations |
|---|---|---|---|
| 1 | 3.61 | 3.07 | 2.99 |
| 2 | 3.89 | 4.32 | 4.27 |
| 3 | 3.47 | 2.95 | 2.94 |
| 4 | 4.23 | 4.75 | 4.83 |
| 5 | 2.81 | 3.01 | 2.91 |
| 6 | 3.86 | 2.25 | 2.27 |
| 7 | 2.58 | 2.32 | 2.25 |
| 8 | 4.57 | 2.61 | 2.53 |
| 9 | 7.76 | 4.47 | 3.30 |
| 10 | 2.35 | 1.83 | 1.83 |
| 11 | 3.68 | 2.82 | 2.25 |
| 12 | 3.33 | 2.32 | 2.32 |
| 13 | 6.65 | 7.80 | 7.81 |
| 14 | 3.39 | 2.27 | 2.28 |
| 15 | 7.69 | 3.45 | 3.45 |
| 16 | 2.61 | 3.14 | 3.10 |
| 17 | 2.53 | 1.53 | 1.54 |
| 18 | 3.17 | 1.63 | 1.66 |
| 19 | 3.39 | 1.78 | 1.81 |
| 20 | 3.19 | 2.32 | 3.98 |
| 21 | 4.19 | 2.02 | 1.99 |
| 22 | 2.49 | 2.03 | 1.95 |
| 23 | 2.56 | 2.00 | 2.08 |
| 24 | 2.05 | 1.60 | 3.16 |
| Average | 3.75 | 2.85 | 2.90 |

About the valuation period, we are sorry for the unclear expression. The NSE, RMSE and PDF-based skill score are calculated from the same period 1979 to 2013. Because we want to test how well is the new data set for entire CTM rather than individual sites. Thus we used quantile function which is allowed to test the data in different periods and different time scales. The mean, maximum and minimum values are the basic indicants. In order to investigate the temperature range, different quantiles are used to represent the distribution. We revised this part in a more clear expression (**P7 L9-11**).

We agree that the temperature varies significant in different seasons and

different times of the day due to the complex topography. For example, in the winter night, the lapse rate is possible reverse (local inversion) from the bottom of valley to the high mountain due to the 'cold lake' (Gerlitz 2014). We added more discussion on this aspect (**Section 4.5 in the revision**). Unfortunately, we did not have sub-day observations to validate. We have used the best we have. In order to identify the limitations for end-users, we tested the seasonal bias using the 24 sites. Table 4 shows the RMSE of seasonal mean temperatures between original ERA-Interim and corrected temperatures for all sites. The RMSE for spring ranges from 0.26 to 4.22 °C with an average of 1.24 °C. The performance for summer and autumn is similar with around 1.4 ° C RMSE. Winter has the largest average RMSE (2.96 °C) over the year. Different stations show significant different performances. For example, station No. 13 shows the largest RMSE for winter while smallest RMSE for summer over all sites. Station No. 9 show the opposite performances that summer has the largest RMSE (5.47 °C) while winter has the smallest RMSE (2.32 °C). This further illustrates that the complex terrain of the CTM leads to the complexity and diversity of the climate. The Supplement 1 shows the RMSE between original ERA-Interim and corrected temperatures at 24 sites for 12 months, which could help the potential users check the bias of the data set. However, in general, the warmer season (May to September) is much better than colder months (**P10 L5-15**).

Table 4 RMSEs between the seasonal ERA-Interim and corrected temperatures for the 24 sites.

| ID | Spring | Summer | Autumn | Winter |
|---|---|---|---|---|
| 1 | 1.33 | 0.67 | 1.61 | 3.70 |
| 2 | 1.99 | 2.63 | 3.18 | 5.32 |
| 3 | 0.57 | 0.66 | 1.17 | 4.24 |
| 4 | 1.56 | 0.89 | 2.47 | 7.69 |
| 5 | 1.38 | 1.79 | 1.49 | 4.02 |
| 6 | 0.47 | 1.63 | 0.96 | 1.16 |

| | | | | |
|---|---|---|---|---|
| 7 | 0.89 | 1.42 | 1.78 | 0.64 |
| 8 | 0.40 | 1.88 | 0.60 | 3.14 |
| 9 | 4.22 | 5.47 | 3.65 | 2.32 |
| 10 | 0.84 | 1.62 | 0.85 | 0.91 |
| 11 | 1.78 | 1.28 | 2.07 | 3.61 |
| 12 | 1.02 | 0.78 | 0.52 | 1.84 |
| 13 | 3.22 | 0.42 | 3.23 | 12.80 |
| 14 | 0.54 | 1.00 | 0.69 | 2.84 |
| 15 | 2.04 | 0.95 | 0.95 | 2.67 |
| 16 | 0.83 | 2.76 | 2.38 | 3.32 |
| 17 | 0.51 | 1.20 | 0.74 | 0.71 |
| 18 | 1.03 | 0.85 | 0.49 | 0.72 |
| 19 | 1.36 | 0.71 | 1.02 | 0.61 |
| 20 | 1.11 | 1.65 | 1.05 | 1.77 |
| 21 | 0.26 | 0.58 | 0.58 | 1.57 |
| 22 | 0.63 | 0.62 | 0.89 | 2.63 |
| 23 | 0.48 | 1.70 | 1.54 | 1.25 |
| 24 | 1.24 | 0.71 | 0.72 | 1.59 |
| Average | 1.24 | 1.41 | 1.44 | 2.96 |

It is true that the lapse rates are used firstly at month scale. The environmental lapse rate is quite different in the free-air atmosphere. The authors have tested the daily and 3-hourly lapse rates in the German and Swiss Alps. The results showed that in general the ERA-Interim internal lapse rates could capture the variability of observed lapse rates, although the performances were different for different grid cell. According to reviewer's suggestion, we added the evaluation of the lapse rates between ERA-Interim and observations (Figure 1). In previous studies (Gao et al., 2012, 2017), the observed lapse rate was calculated from 2 or 3 sites within a same ERA-Interim grid. And then, the observed lapse rate was compared with ERA-Interim internal lapse rate. Unfortunately, the sparse stations cannot support to do this calculation. Thus, we investigate the lapse rate based on the temperature and elevation information from all 24 sites using the linear regression approach for 1979 to 2013. Because the sites elevation ranges from 35 to 2458 m, thus for convenience, the ERA-Interim lapse rate was calculated using the temperature and geopotential height at 925 hPa and 700 hPa levels. The geopotential

height at these two pressure levels range from around 150 m to 3000 m, which is close to the sites' elevation. Thus, the monthly lapse rate for observation and ERA-Interim from 1979 to 2013 was calculated, respectively. Figure 1 shows the temporal variation of monthly lapse rates. In general, the ERA-Interim has a higher temperature gradient than observation for the whole year. However, ERA-Interim captures the variability of observed lapse rate very well, especially in the warmer months (May to August). The inter-monthly variability of observed lapse rate is much higher than ERA-Interim, especially from September to January. The temperature gradient decreases significant from September, which represents the transition month from warm to cold climate regime. The temperature gradient increases significant from March, which represents the climate regime transfers from cold to warm conditions. Table 5 shows the monthly lapse rates over all sites in 1979-2013. The lapse rate differences are small (less than 0.5 °C km$^{-1}$ ) from May to August, while the differences are larger than 1 °C km$^{-1}$ from September to December as well as January (**Section 4.2, P8 L29-31, P9 L1-20**).

[Figure]

Figure 1 Boxplots of monthly lapse rates for observation and ERA-Interim

($\Gamma_{700\_925}$). Thick horizontal linesin boxes show the median values. Boxes indicate the inner-quantile range (25% to 75 %) and the whiskers show the full range of the values.

Table 5 Monthly lapse rate (°C km$^{-1}$) over the 24 sites in 1979-2013.

| Month | observation | $\Gamma_{700\_925}$ |
|---|---|---|
| January | -2.79 | -4.00 |
| February | -4.01 | -4.81 |
| March | -5.42 | -5.96 |
| April | -6.14 | -6.90 |
| May | -6.92 | -7.35 |
| June | -7.55 | -7.52 |
| July | -7.48 | -7.49 |
| August | -6.95 | -7.40 |
| September | -5.93 | -7.10 |
| October | -4.86 | -6.27 |
| November | -3.94 | -4.95 |
| December | -2.88 | -3.88 |

It is true that for a couple of few sites, the data set only show a little bit better or even worse than the original ERA-Interim. The reason is complicated. For example, it is possible that the winter or summer lapse rates are not well simulated by ERA-Interim, especially in the deep valley. We tried to revise this part to make it clearer to follow. Meanwhile, we clarify the strengths and limitations of the data set for the potential users (**Section 4.2, P8 L29-31, P9 L1-20**).

4. Application of the data set The authors show very general characteristics of the dataset, such as mean, minimum and maximum temperatures, in section 4.3. Most applications, which are mentioned in the introduction, however require both, high resolution temperature and precipitation data. I feel that the potential of such a data set should be better illustrated by showing its unique features. Does the high resolution data set e.g. reproduces elevation depending warming in the Tienshan mountains? (see e.g. (Gerlitz et al. 2014)). Are spatial and seasonal variations of the dirurnal temperature range well

captured (Sun et al. 2018; Shekhar et al. 2018)? Such potential applications could be included without much effort and will certainly illustrate the potential of the data set, which stands out due to its spatial AND temporal resolution.

--Answer: Thanks a lot for the comments. The reviewer raised a very important issue on the ability of new data set on the warming trends. We compared the warming trends of observation against the original ERA-Interim and the correction temperatures over the 24 sites in 1979-2013 (Figure 2). Furthermore, we added more analysis on the maximum temperature (Tmax), minimum temperature (Tmin) and diurnal temperature range (DTR) in the revision according to the comments (**Section 4.5 P12, P13 L1-18**). The original ERA-Interim underestimated significant (around 2°C) the observations. However, the corrections overestimated around 1 °C. The annual warming trend with an increase rate of 0.420 °C 10a$^{-1}$ for observation. Generally, the original ERA-Interim and correction temperatures captures the warming trend very well with the rate of 0.378 and 0.349 °C 10a$^{-1}$, respectively. Table 6 shows the trends for seasonal temperatures over 24 sites in 1979-2013. Spring has the largest positive trend with the rate of 0.664 °C 10a$^{-1}$. The original ERA-Interim and correction temperatures captured the warming trends for spring quite well with the rate of 0.659 and 0.638 °C 10a$^{-1}$, respectively. The correction temperatures have the better performance than the original ERA-Interim for summer trend. However, the ERA-Interim and corrections both underestimate the trend with almost the same rate for autumn trend. Unfortunately, the slight positive warming trend for winter is not captured by the original ERA-Interim and correction temperatures. These two data show the similar negative trends.

[Figure]

Figure 2 Temporal variations of annual temperatures for observation, original ERA-Interim and the correction temperatures over the 24 sites in 1979-2013.

Table 6 Trends (°C 10a$^{-1}$) of annual and seasonal temperatures over the 24 sites in 1979-2013.

|  | Annual | Spring | Summer | Autumn | Winter |
|---|---|---|---|---|---|
| observation | 0.420 | 0.664 | 0.432 | 0.532 | 0.018 |
| ERA-Interim | 0.378 | 0.659 | 0.530 | 0.448 | -0.153 |
| Correction | 0.349 | 0.638 | 0.478 | 0.443 | -0.195 |

Figure 3 shows the temporal variations of Tmax over the 24 sites in 1979-2013. The bias of ERA-Interim is around 4 °C compared to observations. The corrections have bias less than 2 °C. The variations are in consistent with the similar warming trend. Table 7 shows the trends for seasonal Tmax over the 24 sites in 1979-2013. In general, the original ERA-Interim and corrections capture the warming trend quite well (~0.370 °C 10a$^{-1}$). Observation has the largest positive trend in spring with the rate of 0.693 °C 10a$^{-1}$ followed by the autumn (0.528 °C 10a$^{-1}$). The warming trends are slight overestimated by ERA-Interim and corrections for summer. The original ERA-Interim and corrections capture the negative trend for winter, but with a higher magnitude than observation.

[Figure]

Figure 3 Temporal variations of Tmax from observation, original ERA-Interim and correction temperatures over the 24 sites in 1979-2013.

Table 7 Trends (°C 10a$^{-1}$) of annual and seasonal Tmax over the 24 sites in 1979-2013.

|  | Annual | Spring | Summer | Autumn | Winter |
|---|---|---|---|---|---|
| Observation | 0.370 | 0.693 | 0.397 | 0.528 | -0.176 |
| ERA-Interim | 0.367 | 0.741 | 0.468 | 0.478 | -0.262 |
| Correction | 0.379 | 0.767 | 0.461 | 0.507 | -0.261 |

Figure 4 demonstrates the temporal variations of Tmin over the 24 sites in 1979-2013. The original ERA-Interim agrees with observations very well with less than 1 °C. The corrections have bias around 2 °C compared to observations. The original ERA-Interim and corrections underestimate the observed warming trend. Table 8 shows the specific values on the trends for seasonal Tmin over 24 sites in 1979-2013. In general, the original ERA-Interim and corrections capture the warming trends for spring, summer and autumn in lower rates, especially for spring and autumn (Table 8). Observation has the largest positive trend in spring with the rate of 0.700 °C 10a$^{-1}$ followed by the autumn (0.661 °C 10a$^{-1}$). The observed warming trend for winter is positive with the rate of 0.209 °C 10a$^{-1}$. However, the ERA-Interim and corrections did not capture the positive trend.

[Figure]

Figure 4 Temporal variations of Tmin from observation, original ERA-Interim and correction temperatures over the 24 sites in 1979-2013.

Table 8 Trends (°C 10a$^{-1}$) of annual and seasonal Tmin over the 24 sites in 1979-2013.

|  | Annual | Spring | Summer | Autumn | Winter |
|---|---|---|---|---|---|
| observation | 0.547 | 0.700 | 0.578 | 0.661 | 0.209 |
| ERA-Interim | 0.338 | 0.479 | 0.519 | 0.409 | -0.084 |
| Correction | 0.344 | 0.493 | 0.505 | 0.439 | -0.093 |

Figure 5 demonstrates the temporal variations of DTR over the 24 sites in 1979-2013. The original ERA-Interim has a more than 3 °C DTR bias compared to observations. The corrections reduce the DTR bias insignificant. The original ERA-Interim and corrections did not capture the significant decreasing trend of DTR. Table 9 shows the specific values on the trends for seasonal DTR over the 24 sites in 1979-2013. The decreasing trends are observed for annual and four seasonal DTR. Winter has the largest decreasing rate with the value of -0.384 °C 10a$^{-1}$. Spring has the insignificant decreasing trend (-0.001 °C 10a$^{-1}$), which may result from the significant increasing rate of Tmax. The original ERA-Interim and corrections capture the decreasing trends for summer and winter with smaller rates. However, they capture the opposite trends for spring and autumn, especially for spring (Table 9). The main reason is that the increasing rates for spring for Tmin are significant underestimated

by the original ERA-Interim and corrections (Table 8).

[Figure]

Figure 5 Temporal variations of DTR from observation, original ERA-Interim
and correction temperatures over the 24 sites in 1979-2013.

Table 9 Trends (°C 10a$^{-1}$) of annual and seasonal DTR over the 24 sites in
1979-2013.

|  | Annual | Spring | Summer | Autumn | Winter |
|---|---|---|---|---|---|
| observation | -0.177 | -0.001 | -0.181 | -0.132 | -0.384 |
| ERA-Interim | 0.029 | 0.262 | -0.052 | 0.069 | -0.178 |
| Correction | 0.036 | 0.274 | -0.044 | 0.068 | -0.168 |

We would like to emphasize that we did not compare the whole CTM DTR with
the 24 observations. The analysis on the Tmax, Tmin and DTR show that the
corrections can capture the annual trend generally, although it is not well on
the seasonal scale. But it is ture that we need more observations to validate
the performance of new data set on DTR and spatial variations at local scales.
Meanwhile, we are collecting local observations at special basins (for example
Kaidu river basin) where are more interesting for researchers to validate the
new data set (**Section 4.5 P12, P13 L1-18**).

5. Data Availability The structure of the data set seems to be a bit unintuitive to
me. Wouldn't it be an option to provide the NCDF files for each year and for the
entire domain? This would simplify the usaga of the data set, particularly for

users who download the data set via batch scripts.

--Answer: Thanks a lot for the comments. Yes, the data set is not unintuitive to users. Because of the large data, we have to divide it into small parts with a limited points and short time series. We tried to put all points together for a single year in a signal NetCDF file, but it was more than 5 GB. It cannot be open by a computer with limited memory. I tried many ways, but it takes so much time to wait for opening the file. The software like Matlab cannot process and analyze the data because it always says out of memory. Thus, we prefer to provide the small part and the potential users can download the data according to the coordinates of study area, rather than download the whole data points. But for sure, we are working on the version 2.0, which is friendlier for users. The accessibility of data set also will be improved in the version 2.0 (**P15 L12-18**).

Gerlitz, L., 2014: Using fuzzified regression trees for statistical downscaling and regionalization of near surface temperatures in complex terrain. Theor Appl Climatol, 122, 337–352, doi:10.1007/s00704-014-1285-x.

Gerlitz, L, O. Conrad, A. Thomas, and J. Böhner, 2014: Warming patterns over the Tibetan Plateau and adjacent lowlands derived from elevation- and bias-corrected ERA-Interim data. Climate Research, 58, 235–246, doi:10.3354/cr01193.

Shekhar, M. S., U. Devi, S. K. Dash, G. P. Singh, and A. Singh, 2018: Variability of Diurnal Temperature Range During Winter Over Western Himalaya: Range- and Altitude-Wise Study. Pure Appl. Geophys., 1–13, doi:10.1007/s00024-018-1845-6.

Sun, X., and Coauthors, 2018: Global diurnal to in-temperature range (DTR) changes since 1901. Clim Dyn, 1–14, doi:10.1007/s00382-018-4329-6.

References:

Gao, L., Hao, L., and Chen, X. W.: Evaluation of ERA-Interim monthly temperature data over the Tibetan Plateau, J. Mount. Sci., 11(5), 1154-1168, doi:10.1007/s11629-014-3013-5, 2014.

Gao, L., Bernhardt, M., Schulz, K., and Chen, X. W.: Elevation correction of ERA-Interim temperature data in the Tibetan Plateau, Int. J. Climatol., 37, 3540–3552, doi: 10.1002/joc.4935, 2017.

Gerlitz, L., 2014: Using fuzzified regression trees for statistical downscaling and regionalization of near surface temperatures in complex terrain. Theor Appl Climatol, 122, 337–352, doi:10.1007/s00704-014-1285-x.

Gerlitz, L, O. Conrad, A. Thomas, and J. Böhner, 2014: Warming patterns over the Tibetan Plateau and adjacent lowlands derived from elevation- and bias-corrected ERA-Interim data. Climate Research, 58, 235–246, doi:10.3354/cr01193.

---

## Author Comment (AC2) · 16 Sep 2018

**Response to Reviewer 2**

We would like to thank the anonymous referee for reviewing our manuscript. These constructive comments are very important for us to improve the present manuscript. In the following, we address all comments point-by-point according to the comments. All revisions are highlighted in the context.

The authors present a high-resolution (1km, 6h) air temperature data set for the Chinese Tianshan Mountains from 1979 to 2016 based on a downscaling method. This topic is quite interesting and the data set would be useful for the potential end-users who focus on the alpine climate and cryosphere issues in the Tianshan Mountains. In general, this paper is well-written for most parts. However, a major revision is needed before it is published in ESSD.

General Comments:

1.  The mentioned downscaling method in the paper has been validated in the Alps Mountains and the Tibet Plateau. However, the relative references (Gao et al., 2012, 2017) named this method as "elevation correction" rather than "downscaling". What is the difference between these two methods or terms? For me, they are the same. Thus, which one is more appropriate?

-Answer: Thanks a lot for the comments. In this study we used the term of "downscaling" to emphasize the spatial resolution enhances from 0.25° to 1 km grid. But the reviewer is right that the correction has the same meaning with downscaling in this study. The elevation correction can be considered a form of downscaling. The reviewer 1 (Dr. Gerlitz) also pointed out this issue. He thought that correction is more intuitive because we only used the single factor (elevation) rather than local-scale processes. We revised this term in the revision version.

2.  24 meteorological stations are not enough for validation for such a large

area (more than 80000 points).   Is there any other data resources could be used for further validation?

-Answer: Thanks a lot for the comments. Validation is the necessary process for a new data set. Unfortunately, the observation sites in the CTM are really sparse. This is the motivation we want to product a high-resolution spatial continuous and long-term data set for the CTM. There are around 760 meteorological sites with good quality (daily, more than 40 years) can be used over the whole China. Among them around 35 sites in Xinjiang Province with long-term records can be used. For the CTM, only 24 sites are available for validation. This is the best data resource we have. For sure, there are some global grid data sets which cover the CTM. However, most of them are produced by interpolated or modeled (e.g. CRUTEM3, E-OBS). The remote sensing has shorter time series and large bias. We believe that the observation from the meteorological stations is the best data set for validation. It is unscientific to validate the new data set using the data that contains bias. Although, 24 sites are not enough for validation, these sites can generally test the quality of the new data set. We welcome the potential users to evaluate the new data set based on different data resources that we do not have. More assessments can help us improve the product.

3.   The authors pointed out that about 24% of RMSE was reduced by the downscaling method compared to the original ERA-Interim. Is it good enough? How to evaluate the data set (or any reference/standard) is good enough for end-users?

-Answer: Thanks a lot for the comments. To be frank, this is a difficult question to answer. Normally, there is no standard to measure the significance of error reduction. However, we thought that 24% of RMSE (around 1°C ) is quite good for the CTM. For mountainous, the original ERA-Interim always has a large

bias more than 3°C, for example 3.7°C in the Tibetan Plateau (Gao et al., 2014). Gao et al. (2017) used the similar method reduced 62% of RMSE for ERA-Interim (from 4.31 to 1.64°C) in the Tibetan Plateau. But in that study, 80 meteorological stations are used for validation. This is another potential cause for the "insignificant" bias reduction in the present study. Thus, we need more observations and other data resources as well as other users to evaluate whether the new data is good enough.

4. For my understanding, the downscaling method is mainly based on the elevation (DEM). Is it possible to get higher resolution data set if we use the 100 m DEM? The ERA-Interim product provides 3-hourly forecast data. Thus, is it possible to obtain 3-hourly data set for whole Tianshan Mountains?

-Answer: Thanks a lot for the comments. The reviewer raised a very interesting question. In fact, at the very beginning of this study, we planned to produce a 100m 3-hourly temperature data set for the CTM. However, we met two challenges: 1) China is not the member country of ECMWF, which means we cannot gain the 3-hourly forecast data in a direct way. We only can download the 6-hourly analysis data from the public data set archive; 2) 100 m resolution means the total number of grid reaches more than 8 millions. For an ordinary computer even a computer workstation, it is too large to process. For the end-users, such a large amount of data is not convenient. We plan to produce 100m 3-hourly data sets in the future but only for selected area such as basins or valleys that the end-user interested in.

5. How to evaluate the lapse rate is correct or appropriate for the downscaling? The lapse rate varies significant in different topographical situations and time period.

-Answer: Thanks a lot for the comments. This is an important issue. The lapse rate has a large spatial-temporal variability in the mountain areas. The

reviewer 1 also pointed this question out. We added the evaluation of the lapse rates between ERA-Interim and observations in the section 4.2 in the revision (Figure 1). Figure 1 shows the temporal variation of monthly lapse rates for observation and ERA-Interim ($\Gamma_{700\_925}$) over the 24 sites. In general, the ERA-Interim has a higher temperature gradient than observation. However, ERA-Interim captures the variability of observed lapse rate very well, especially in the warmer months (May to August). The inter-monthly variability of observed lapse rate is much higher than ERA-Interim, especially from September to January. Table 1 shows the monthly lapse rates over all sites in 1979-2013. The lapse rate differences are small (less than 0.5 °C km$^{-1}$) from May to August, while the differences are larger than 1 °C km$^{-1}$ from September to December and January. More details please see the section 4.2 in the revision (**P8 L29-31, P9 L1-20**).

[Figure]

Figure 1 Boxplots of monthly lapse rates for observation and ERA-Interim ($\Gamma_{700\_925}$). Thick horizontal linesin boxes show the median values. Boxes indicate the inner-quantile range (25% to 75 %) and the whiskers show the full range of the values.

Table 1 Monthly lapse rate (°C km$^{-1}$) over all sites in 1979-2013.

| Month | observation | $\Gamma_{700\_925}$ |
|---|---|---|
| January | -2.79 | -4.00 |
| February | -4.01 | -4.81 |
| March | -5.42 | -5.96 |
| April | -6.14 | -6.90 |
| May | -6.92 | -7.35 |
| June | -7.55 | -7.52 |
| July | -7.48 | -7.49 |
| August | -6.95 | -7.40 |
| September | -5.93 | -7.10 |
| October | -4.86 | -6.27 |
| November | -3.94 | -4.95 |
| December | -2.88 | -3.88 |

6. Precipitation is another basic and important variable for climate and environmental models. Can you produce any high resolution precipitation data set using some similar methods for this region?

-Answer: Thanks a lot for the comments. This is another interesting issue. Hundreds methods/models for precipitation downscaling were developed in the past two decades. However, at present, there is no universal method. Meanwhile, precipitation is more complex and difficult to downscale for a finer grid, especially independent of observations. To our knowledge, PRISM (Precipitation-elevation Regression on Independent Slope Model) is one of the possible methods to downscale the coarse grids to a finer grid. PRISM takes the elevation and other microtopographic factors (slope, aspect) into account. Until now, we did not test this method in the CTM. But it is our future research plan. Surely, PRISM should be adjusted and further developed for the CTM because the snow is the main form of precipitation in winter.

Specific comments:

1. Although the authors listed many references about the downscaling method, I believe it is necessary to clarify the method specific for the readers

-Answer: Thanks for pointing this out. We added more information on the correction methods in the section 3.3. We also gave an example to calculate the lapse rate (**P6 L2-11**).

2. The downscaling method is more appropriate named "elevation correction". Since only elevation is involved. The conventional circulation variables such as wind, sea level pressure, humidity are not considered in the downscaling method.

-Answer: Thanks for pointing this out. Yes, we agree that elevation correction is more appropriate for this study. We have changed it for the whole context in the revision.

3. The data set is not friendly to download and use. The data set is divided into so many sub-files. Is it possible to find a more easy way for users?

-Answer: Thanks. Other reviewers also pointed this problem out. We also realize that the data set is not very friendly. We have tried many ways to make it easier for end-user. For example, we put all points together for a single year in a signal NetCDF file, but it was more than 5 GB. A normal desktop cannot read it. The Matlab (we process the data in Matlab) always says out of memory. Thus, we prefer to provide the small parts. The advantage is that the potential users can download the data points according to the coordinates of study area, rather than download the whole data points. We are working on the version 2.0, which is friendlier to users. The accessibility of data set also will be improved in the version 2.0 (**P15 L12-18**).

4. The downscaled data at some sites are worse than the original ERA-Interim data. Why? The authors should discuss this issue. It is very important because only 24 sites are available for validation.

-Answer: Thanks a lot for the comments. It is true that the elevation correction method did not work very well in some sites. We thought that the micro-topographical features such as aspect and slope of the mountain are the main reason. For example, for station No. 16 (Baicheng) located in the valley, the lapse rate might changes sharply even inverse in the cold winter. We added some discussion in the revision (**P14 L19-23, 28-29**).

5. If someone plans to run a hydrology model in a small catchment in the Tianshan Mountains. How to adjust the data set points to match the model grids?

-Answer: Thanks a lot for the comments. Firstly, the end-user should confirm the boundary (coordinates of the four directions) of the basin. Secondly, confirm the center points of hydrological model grids, for example 3 km grid. Each grid has a center point with the coordinate. Third, download all the grids (818126) information (not the data, only a small txt file) from the data set. This file contains the coordinates of all points and their elevation. Last, the data point can be selected by calculating the distance (define a threshold according to the hydrology model, for example 3 km) from the data point to the model point. There is a more intuitive way using ArcMap software. Firstly, create two grids layers in ArcMap, and then overlay or interact two layers. The overlap part is the matching points for data points and model points. Then, export the coordinates of matching grids from ArcMap. According to the ID of data points, download the relative data from PANGAEA for the time series the users want.

6. I found that the data amount is around 187G. How to process such large data set? What is software or platform to process it? Maybe, the authors could provide some codes for data processing.

-Answer: Thanks for the question. We have spent much time on the data processing and calculation. Matlab is the tool we used. Because of the large

data amount, we used the VSC-3 super computer system in Vienna of Austria. It took almost one month to finish the calculation. But now, the end-users can process and analyze data on the laptop or desktop because we divided the grids and time series into smaller ones (the amount of single file is less than 2 GB). The Matlab code is simple to follow. We could provide it if some end-users need it.

7. I am not sure the data set could capture the temperature changes in the micro-topography since the original data is 0.25 degree. The slope and aspect of mountains also affect the temperature significant, especially in the night.

-Answer: Thanks a lot for the comments. The reviewer raised a very important issue. The original spatial resolution of ERA-Interim is reduced Gaussian grid N128 (around 0.75 degree). In general, ERA-Interim has a relative small large-scale bias. However, we do not think it can capture the temperature changes in the micro-topography. Other reviewers also mentioned this question. We added more analysis on the maximum temperature, minimum temperature and diurnal temperature ranges as well as the warming trends. This analysis could generally illustrate the ability of the new data set on the temperature changes. In the future, we plan to adjust the temperature according to the micro-topography features, especially for local scales such as basins. We added some discussion on this issue in the revision (**Section 4.5 P12, P13 L1-18**).

8. Can other temperature downscaling methods be used for the high-resolution data set? And why?

-Answer: Thanks a lot for the comments. This is an interesting open question. The downscaling based on the lapse rate is the most common used method for temperature. However, most cases used the fixed lapse rate (-6.0 or -6.5 °C km$^{-1}$) or monthly lapse rate from Kunkel (1989). Fewer studies focused on the

lapse rate variability. We do not think the conventional method could be used for high-resolution data set, especially for the high mountain areas. These methods rely on a high density of observations. However, the sites are very sparse in the high mountain areas. The present method is independent of observations, which could be extended to any other high mountain areas over the world.

9. Some expression and description of language is not clear. A native speaker would be helpful for the improvement of readability for the whole context.

-Answer: Thanks a lot for the comments. It is true that there are some language problems. The English expression in the revision is corrected by the Elsevier publishing group (https://webshop.elsevier.com/)

References:

Gao, L., Hao, L., and Chen, X. W.: Evaluation of ERA-Interim monthly temperature data over the Tibetan Plateau, J. Mount. Sci., 11(5), 1154-1168, doi:10.1007/s11629-014-3013-5, 2014.

Gao, L., Bernhardt, M., Schulz, K., and Chen, X. W.: Elevation correction of ERA-Interim temperature data in the Tibetan Plateau, Int. J. Climatol., 37, 3540–3552, doi: 10.1002/joc.4935, 2017.

Kunkel, E. K.: Simple Procedures for Extrapolation of Humidity Variables in the Mountainous Western United States, J. Climate, 2, 656–669, 1989.

---

## Author Comment (AC3) · 16 Sep 2018

**Response to Reviewer 3**

We would like to thank the anonymous referee for reviewing our paper. These constructive comments are very important for improving the present version manuscript. In the following, we address all comments point-by-point according to the comments. All revisions are highlighted in the context.

This is an interesting piece of research that are focusing on downscaling temperature data for one mountainous region in China. The study is clearly written and the data/method is solid. I recommend a "minor revision" for this paper. Below is my comments and questions.

1. Needs a short introduction of the mechanism of the method mentioned in line 31, page 2.

-Answer: Thanks a lot for the comment. We agree that the method should be presented in more detail. We added more information in this part and also on the correction method in the section 3.3. An example on the internal lapse rate scheme was presented (**P6 L2-11**).

2. The labels for stations in figure 1 are not very clear, please use another color.

-Answer: Thanks a lot for pointing this out. We made the station mark and the labels to yellow color in Figure 1. It is clearer than the old one.

[Figure]

Figure 1: Location of the 24 meteorological stations (triangle) and ERA-Interim 0.25°×0.25° grid points (dot). The elevation ranges from -161 m to 7100 m a.s.l., with a DEM resolution of 1km.

3. In figure 2 and 4, why Nos.10, 24, 9 and 15 stations are chosen for comparison? Are they the most representative ones?

-Answer: Thanks a lot for the comments. Yes, in order to save the space of the paper, we selected four representative stations for examples. These stations represent different sub-climate regimes and topography situations in the north slope of the CTM, south slope of the CTM, eastern CTM and western CTM.

4. Although the author mentioned that the temporal resolution of the new dataset is 6 hours, non of the analysis is focused on the temporal variations of the temperature. Needs at least some analysis to address the temporal accuracy of the new dataset.

-Answer: Thanks a lot for the comments. The reviewer raises a very important issue. We added the analysis on the temporal variability of lapse rate in section 4.2 (**P8 L29-31, P9 L1-20**). We also compared the warming trends of observation against the original ERA-Interim and the correction temperatures

over 24 sites in 1979-2013 in section 4.3. Furthermore, we added more analysis on the maximum temperature (Tmax), minimum temperature (Tmin) and diurnal temperature range (DTR) in section 4.5 (**P12, P13 L1-18**). These analyses address the temporal accuracy of the new dataset and also illustrate the limitations to potential users.

5. Page 9, line 23, delete "there are".

-Answer: Thanks a lot for pointing this out. We revised it.

6. How the comparison error statistics are sensitive to the "downscaling resolution" you have chosen? Why the final product is set to 1 km grids?

-Answer: Thanks a lot for the comments. We selected the most common used statistics for error comparison. We compared the RMSEs of original and corrected ERA-Interim temperature using surrounding 9 grids as well as directly using station elevations. The RMSE differences between two approaches are small for all most of the sites (averaged RMSE only 0.05 °C). Only two stations (Nos. 20 and 24) show large differences. It is possible relate to the large elevation differences among 9 DEM grids. But the comparison illustrate the statistics are not sensitive to downscaling resolution in general.

Table 1 RMSE of original and downscaled ERA-Interim temperature using 9 grids as well as directly with station elevations.

| ID | original ERA-Interim | corrected based on 9 DEM grids | corrected directly based on station elevations |
|----|----|----|----|
| 1 | 3.61 | 3.07 | 2.99 |
| 2 | 3.89 | 4.32 | 4.27 |
| 3 | 3.47 | 2.95 | 2.94 |
| 4 | 4.23 | 4.75 | 4.83 |
| 5 | 2.81 | 3.01 | 2.91 |
| 6 | 3.86 | 2.25 | 2.27 |
| 7 | 2.58 | 2.32 | 2.25 |
| 8 | 4.57 | 2.61 | 2.53 |

| | | | |
|---|---|---|---|
| 9 | 7.76 | 4.47 | 3.30 |
| 10 | 2.35 | 1.83 | 1.83 |
| 11 | 3.68 | 2.82 | 2.25 |
| 12 | 3.33 | 2.32 | 2.32 |
| 13 | 6.65 | 7.80 | 7.81 |
| 14 | 3.39 | 2.27 | 2.28 |
| 15 | 7.69 | 3.45 | 3.45 |
| 16 | 2.61 | 3.14 | 3.10 |
| 17 | 2.53 | 1.53 | 1.54 |
| 18 | 3.17 | 1.63 | 1.66 |
| 19 | 3.39 | 1.78 | 1.81 |
| 20 | 3.19 | 2.32 | 3.98 |
| 21 | 4.19 | 2.02 | 1.99 |
| 22 | 2.49 | 2.03 | 1.95 |
| 23 | 2.56 | 2.00 | 2.08 |
| 24 | 2.05 | 1.60 | 3.16 |
| Average | 3.75 | 2.85 | 2.90 |

7. Have you considered the diurnal changes of the situation in mountainous areas. Since the data you have corrected is at 6 hour resolution. In mountainous area, you may expect temperature inversions during night time.

-Answer: Thanks a lot for the comments. It is a very important issue. Other reviewers also mentioned this topic. We added more analysis on the maximum temperature (Tmax), minimum temperature (Tmin) and diurnal temperature range (DTR) in section 4.5 (**P12, P13 L1-18**). The DTR trend also is explored based on the validation sites.

Figure 2 demonstrates the temporal variations of DTR over the 24 sites in 1979-2013. The original ERA-Interim has a more than 3 °C DTR bias compared to observations. The corrections reduce the DTR bias insignificant. The original ERA-Interim and corrections did not capture the significant decreasing trend of DTR. Table 2 shows the specific values on the trends for seasonal DTR over the 24 sites in 1979-2013. The decreasing trends are observed for annual and four seasonal DTR. Winter has the largest decreasing

rate with the value of -0.384 °C 10a$^{-1}$. Spring has the insignificant decreasing trend (-0.001 °C 10a$^{-1}$). The original ERA-Interim and corrections capture the decreasing trends for summer and winter with smaller rates. However, they capture the opposite trends for spring and autumn, especially for spring (Table 2).

[Figure]

Figure 2 Temporal variations of DTR from observation, original ERA-Interim and correction temperatures over the 24 sites in 1979-2013.

Table 2 Trends (°C 10a$^{-1}$) of annual and seasonal DTR over the 24 sites in 1979-2013.

|  | Annual | Spring | Summer | Autumn | Winter |
|---|---|---|---|---|---|
| observation | -0.177 | -0.001 | -0.181 | -0.132 | -0.384 |
| ERA-Interim | 0.029 | 0.262 | -0.052 | 0.069 | -0.178 |
| Correction | 0.036 | 0.274 | -0.044 | 0.068 | -0.168 |

---

## Author Comment (AC4) · 16 Sep 2018

**Response to the short comments**

We would like to thank Dr. Li Ma for the short comments. These suggestions and comments are useful for improving the present version manuscript. In the following, we address all comments point-by-point according to the comments. All revisions are highlighted in the context.

Producing the high-resolution air temperature dataset in the mountain area is useful for regional climate or hydrological studies. The general target of this study is important. I have some suggestions and comments in below:

1. The method and the used lapse-rates should be more detail described in section 3.3.

-Answer: Thanks a lot for the comment. We agree that the method and the lapse rate should be presented in more detail. We added more information on the correction method in the section 3.3, especially an example on the internal lapse rate scheme (**P6 L2-11**). We also added the analysis on the temporal variability of lapse rate in Section 4.2 (**P8 L29-31, P9 L1-20**).

2. In table 1, is the elevation of the sites as same as the height from the 1 km SRTM DEM grid? it seems not the same (from P9, l18 ., I recommend to list the sites height in 1km DEM.

-Answer: Thanks a lot for the comment. Dr. Ma pointed out a very important issue. The elevations of sites are not the same as the height of 1 km DEM grids. The elevation differences between averaged 9 grids and station elevations are quite small with an average of -8 m (Table 1). Except the No. 9, the rest stations have less than 50 m elevation differences. Thus, the DEM generally matches the station elevations. The averaged 9 DEM grids height is added in the Table S1 (**supplementary Table S1**) in the revision.

Table 1 Elevation of averaged 9 DEM grids and the elevation differences with

station elevation (m).

| ID | averaged 9 DEM grids height | elevation Difference |
|---|---|---|
| 1 | 1305 | 50 |
| 2 | 306 | 14 |
| 3 | 477 | 2 |
| 4 | 467 | -26 |
| 5 | 764 | 30 |
| 6 | 672 | -9 |
| 7 | 1885 | -34 |
| 8 | 893 | 25 |
| 9 | 2004 | -251 |
| 10 | 1101 | 3 |
| 11 | 868 | 5 |
| 12 | 940 | -18 |
| 13 | 2462 | -4 |
| 14 | 1057 | -2 |
| 15 | 11 | 24 |
| 16 | 1221 | 8 |
| 17 | 978 | -2 |
| 18 | 1066 | 33 |
| 19 | 937 | -5 |
| 20 | 1635 | 3 |
| 21 | 433 | 46 |
| 22 | 1814 | -85 |
| 23 | 758 | -21 |
| 24 | 1548 | 20 |
| Average | | **-8** |

3. Addition to comment 2. As the height in DEM grid and site point is different, in P6I13, why do you use averaged 9 grid points to evaluate the downscaled results. In my opinion, since each point in 1 km grid is downscaled according to its DEM height and ERA height, the nest grid point or the nest height grid of 9 point should be used for comparison.

-Answer: Thanks a lot for the comment. The reviewer 1 (Dr. Gerlitz) also pointed out that using the average 9 grid DEM height may lead to a systematic bias since the station elevation does not coincide with the mean elevation of the considered grid cells perfectly. Dr. Gerlitz also suggested correcting the

temperatures directly to site scale. In the validation, the 3*3 grid cells surrounding each station were averaged. This approach was suggested by a referee when the authors evaluated the ERA-Interim temperature over the Tibetan Plateau (Gao et al., 2014). He/she claimed that this way can evaluate the ability of ERA-Interim on different topographies by selecting 3*3 grids with the station located in the center grid. Thus, in this study we took this suggestion. Despite, we found that the elevation differences are very tiny (smaller than 2m) among the 9 grids at 1km *1km grid resolution. Thus, we think this approach does not affect the validation very much (**P7 L12-21**).

4. The lapse rate varies in the different topographical situation and different timestamp like during the nighttime (Li et. al. 2014). The authors should discuss more about the method and results on the diurnal scale.

-Answer: Thanks a lot for the comment. Other reviewer also pointed out this issue. It is necessary to discuss the lapse rate in more detail. We added a new section in section 4.2 to analyze the temporal variability of lapse rates (**P8 L29-31, P9 L1-20**). Until now we do not have the sub-daily observations, we only compared the daily lapse rates between observation and ERA-Interim. The daily lapse rate is aggregated into monthly scale. Because the sites elevation ranges from 35 to 2458 m, thus for convenience, the ERA-Interim lapse rate was calculated using the temperature and geopotential height at 925 hPa and 700 hPa levels. The geopotential height at these two pressure levels range from around 150 m to 3000 m, which is close to the sites' elevation.

Figure 1 shows the temporal variation of monthly lapse rates. In general, the ERA-Interim has a higher temperature gradient than observation for the whole year. However, ERA-Interim captures the variability of observed lapse rate very well, especially in the warmer months (May to August). The inter-monthly variability of observed lapse rate is much higher than ERA-Interim, especially

from September to January. The temperature gradient decreases significant from September, which represents the transition month from warm to cold climate regime. The temperature gradient increases significant from March, which represents the climate regime transfers from cold to warm conditions. Table 2 shows the monthly lapse rates over all sites in 1979-2013. The lapse rate differences are small (less than 0.5 °C km$^{-1}$ ) from May to August, while the differences are larger than 1 °C km$^{-1}$ from September to December and January.

[Figure]

Figure 1 Boxplots of monthly lapse rates for observation and ERA-Interim ($\Gamma_{700\_925}$). Thick horizontal linesin boxes show the median values. Boxes indicate the inner-quantile range (25% to 75 %) and the whiskers show the full range of the values.

Table 2 Monthly lapse rate (°C km$^{-1}$) over all sites in 1979-2013.

| Month | observation | $\Gamma_{700\_925}$ |
| --- | --- | --- |
| January | -2.79 | -4.00 |
| February | -4.01 | -4.81 |
| March | -5.42 | -5.96 |

| | | |
|---|---|---|
| April | -6.14 | -6.90 |
| May | -6.92 | -7.35 |
| June | -7.55 | -7.52 |
| July | -7.48 | -7.49 |
| August | -6.95 | -7.40 |
| September | -5.93 | -7.10 |
| October | -4.86 | -6.27 |
| November | -3.94 | -4.95 |
| December | -2.88 | -3.88 |

5. The authors produce the data from ～25 km to 1 km resolution. The total grid points are 818126, however, only 24 observation stations are used in validation. And in Figure 2-4, the authors only present comparison results at 4 stations, which probably have the best results. In addition, the authors only validate this 6-hourly dataset at a daily scale. To my point, the validation is somehow insufficient. It is not enough to conclude the reliability of this dataset, at least in the current level of discussion of the manuscript. I recommend the authors present some comparison analysis with ERA-Interim on the diurnal scale, and have more validation results with the station observations. Although very limited stations exist in this area, I know the diurnal max. and min. temperatures are provided in CMA station datasets, and these could be used in robust validation.

-Answer: Thanks a lot for pointing this out. It is true that 24 sites are not enough for such a large area. For Figure 2 and 4, the four stations are not the best ones but they were only selected to represent different sub-climate regimes and topography situations in the north slope of the CTM, south slope of the CTM, eastern CTM and western CTM. Prof. Dr. Guoyu Ren in National Climate Center of China and Dr. Aixia Feng in CMA told me (personal communication) that there is a high-density station data set (more than 2000 sites) but only for limited institute. We are trying to collect and apply more observations to validate the new data set. But at present, we have used the best we have. We expect other researchers to validate our product using

different data resources. The validation and application are really welcome (**P15 L12-18**).

In order to illustrate the limitation of the new data set, we added more analysis on the maximum temperature (Tmax), minimum temperature (Tmin) and diurnal temperature range (DTR) in section 4.5. The warming trends for Tmax, Tmin and DRT also are investigated (**P12, P13 L1-18**). For example for DTR:

Figure 2 demonstrates the temporal variations of DTR over the 24 sites in 1979-2013. The original ERA-Interim has a more than 3 °C DTR bias compared to observations. The corrections reduce the DTR bias insignificant. The original ERA-Interim and corrections did not capture the significant decreasing trend of DTR. Table 3 shows the specific values on the trends for seasonal DTR over the 24 sites in 1979-2013. The decreasing trends are observed for annual and four seasonal DTR. Winter has the largest decreasing rate with the value of -0.384 °C 10a$^{-1}$. Spring has the insignificant decreasing trend (-0.001 °C 10a$^{-1}$). The original ERA-Interim and corrections capture the decreasing trends for summer and winter with smaller rates. However, they capture the opposite trends for spring and autumn, especially for spring (Table 3).

[Figure]

Figure 2 Temporal variations of DTR from observation, original ERA-Interim

and correction temperatures over the 24 sites in 1979-2013.

Table 3 Trends (°C 10a$^{-1}$) of annual and seasonal DTR over the 24 sites in 1979-2013.

|  | Annual | Spring | Summer | Autumn | Winter |
|---|---|---|---|---|---|
| observation | -0.177 | -0.001 | -0.181 | -0.132 | -0.384 |
| ERA-Interim | 0.029 | 0.262 | -0.052 | 0.069 | -0.178 |
| Correction | 0.036 | 0.274 | -0.044 | 0.068 | -0.168 |

6. As the dataset provide temperature in 38 years, the authors should show some validation in annual and seasonal scale for this long period. In the spatial scale, I recommend to validate the dataset in each sub regions based on hydrology basin or climate zone or different elevation ranges. Figure 5 and figure 6 can not give much information of any performance skill of this dataset.

-Answer: Thanks a lot for the comments. As we answered for question 5, we are now applying the high-density observations from CMA for further validation. Meanwhile, we discussed with some researchers, for example Dr. Haijun Deng in Fujian Normal University, who did a lot of work in Kaidu river basin and Urumqi river basin. We initially agreed to validate the new data set using some automatic weather observations in a high-temporal resolution (**P13 L14-18**). Figure 5-7 showed the general climatology of the CTM based on the new data set. However, some researchers interested more in maximum and minimum temperatures rather than 1% and 99% quantiles. Thus, the figures for spatial distribution of maximum and minimum temperatures over the CTM are also provided in supplementary (**P11 L26-29, supplementary Figures S1 and S2**).

7. The dataset is very not friendly to use to me. As NetCDF format, I recommend to provide each file for the whole area at each timestamp (or each day, each month, each year) like most Grid datasets did (APHRODITE, TRMM, China meteorological forcing data from CAS. et. al. ). It will be much easier for

regional climate or hydrology studies. Or save as the GRIB format like the reanalysis dataset.

-Answer: Thanks a lot for the suggestions. All reviewers pointed this problem out. We know that the data set is not very friendly at present. We have tried many ways to make it easier for end-user. For example, we put all points together for a single year in a signal NetCDF file, but it was more than 5 GB. A normal desktop cannot read it. The Matlab (we process the data in Matlab) always says out of memory. If we divided into monthly or daily, the numbers of files will be huge. Thus, we prefer to provide the smaller parts with limited points and time series. The advantage is that the potential users can download the data points according to the coordinates of study area. It is not necessary to download the whole data points. We are working on the version 2.0, which is friendlier to users. The accessibility of data set (including data format) also will be improved in the version 2.0(**P15 L12-18**).

Reference:

Li, Y., Z. Zeng, L. Zhao, and S. Piao ( 2015), Spatial patterns of climatological temperature lapse rate in mainland China: A multi–time scale investigation, J. Geophys. Res. Atmos., 120, 2661–2675, doi:10.1002/2014JD022978.